# Tissue shear as a cue for aligning planar polarity in the developing *Drosophila* wing

**Su Ee Tan** ⓘ **& David Strutt** ⓘ ✉

Planar polarity establishment in epithelia requires interpretation of directional tissue-level information at cellular and molecular levels. Mechanical forces exerted during tissue morphogenesis are emerging as crucial tissue-level directional cues, yet the mechanisms by which they regulate planar polarity are poorly understood. Using the *Drosophila* pupal wing, we confirm that tissue stress promotes proximal-distal (PD) planar polarity alignment. Moreover, high tissue stress anisotropy can reduce the rate of accumulation and lower the stability on cell junctions of the core planar polarity protein Frizzled (Fz). Notably, under high tissue stress anisotropy, we see an increased gradient of cell flow, characterised by differential velocities across adjacent cell rows. This promotes core protein turnover at cell-cell contacts parallel to the flow direction, possibly via dissociation of transmembrane complexes by shear forces. We propose that gradients of cell flow play a critical role in establishing and maintaining PD-oriented polarity alignment in the developing pupal wing.

Pattern formation involves symmetry breaking across multiple scales, encompassing the molecular, cellular, and tissue[1]. A simple example is epithelial tissues, which possess a remarkable capacity to collectively orient and align surface structures, including hairs, cilia, and stereocilia. This trait is a shared characteristic conserved across diverse tissues and species and is governed by planar polarity (or planar cell polarity) pathways, which regulate the establishment of polarity within the plane of an epithelium. This process is crucial for the organisation and patterning of tissue, and disruption leads to a range of developmental defects in vertebrates, including humans[2–4]. The core planar polarity pathway (hereafter referred to as the core pathway) is a key molecular pathway required for organisation of planar polarity and is conserved across the animal kingdom[5,6].

The mechanisms behind planar polarisation have been extensively explored in the *Drosophila* wing, where the core pathway establishes the consistent distal orientation of wing hairs (as depicted by 32 hours After Pupal Formation (hAPF) wing in Fig. 1a). Core pathway activity generates asymmetric protein distributions at the adherens junctions that interconnect neighbouring cells. Within the *Drosophila* wing, these complexes become situated along proximal and distal cell interfaces (Fig. 1a'). The core machinery is composed of six core polarity proteins including the seven-pass transmembrane cadherin Flamingo (Fmi; also known as Starry Night [Stan] or Celsr in

vertebrates), responsible for homophilic interactions bridging adjacent cells. Fmi engages with distinct partners on opposing sides of cell contacts. In one cell, Fmi forms complexes containing Frizzled (Fz; Fzd in vertebrates), a seven-pass transmembrane protein, in addition to two cytoplasmic associated proteins, Dishevelled and Diego (Fig. 1a"). In the adjacent cell, Fmi enlists the transmembrane protein Strabismus (Stbm; also known as Van Gogh [Vang] or Vangl in vertebrates), along with the cytoplasmic associated protein Prickle. Loss of core pathway function results in wing trichome initiation initially in the centre of cells followed by trichomes adopting swirling orientations on the wing surface[7,8].

The core proteins are involved in amplifying asymmetries and coordinating polarisation between neighbouring cells, to produce locally aligned polarity. Mosaic clones of *fz* or *stbm* mutants, which disrupt the local organisation of planar polarity, result in misorientation of wing trichomes in the vicinity of the clones[9,10]. This implies that cell polarity is inherently interconnected among neighbouring cells, facilitating the local coordination of polarity independently of tissue-level cues. However, experimental studies along with theoretical models suggest that polarity disruption at a clone boundary can only propagate across several adjacent cells[11,12]. This indicates that intercellular signalling is insufficient to ensure consistent alignment of polarity across the tissue and that additional tissue-level cues must be

School of Biosciences, University of Sheffield, Firth Court, Sheffield, UK. ✉e-mail: d.strutt@sheffield.ac.uk

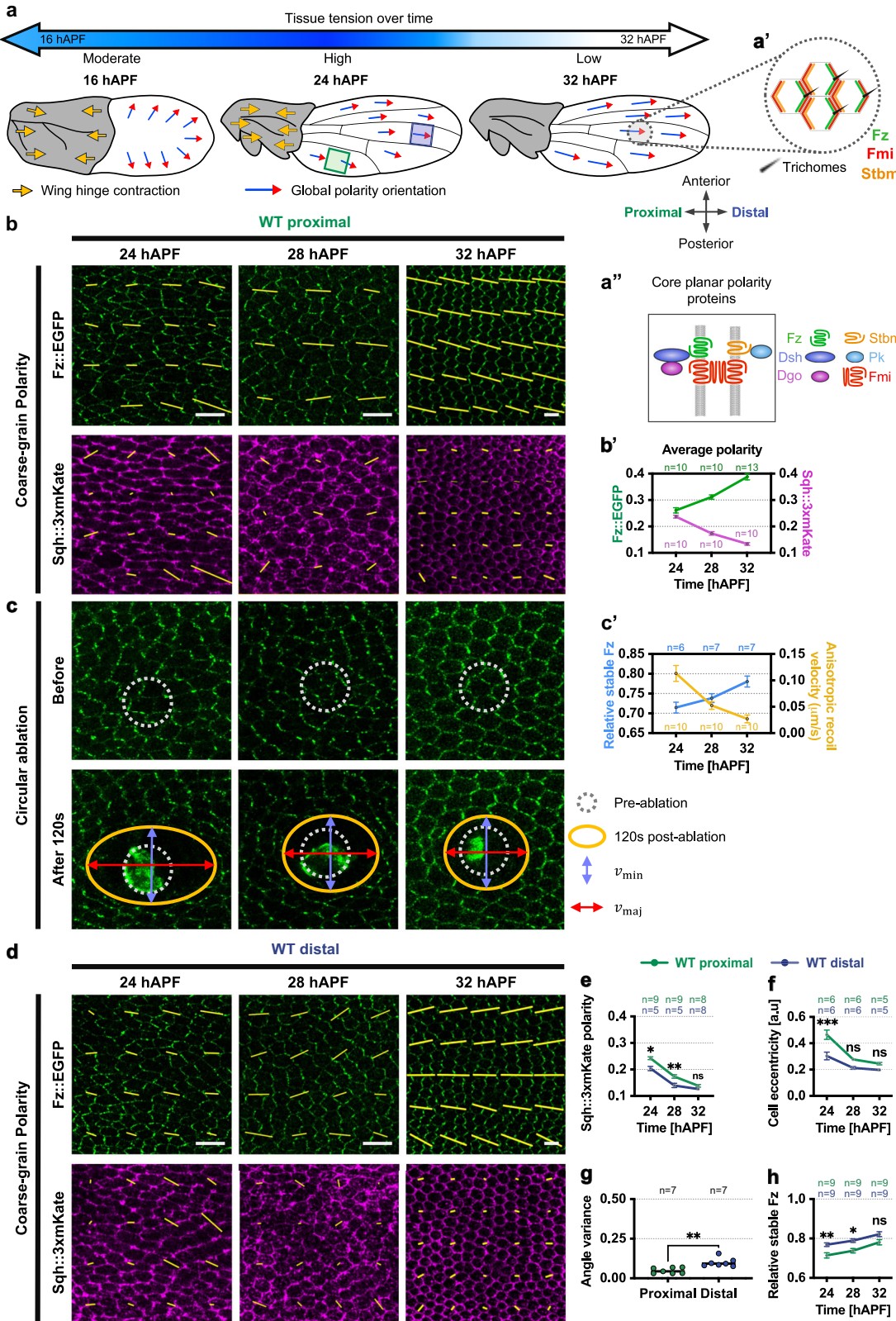

at play to orchestrate planar polarity directionality with respect to the axis of an organ. Pinpointing the precise source of this overarching cue for global polarity coordination has been a long-standing research question[13].

Morphogenetic cues known for their role in tissue patterning during development such as graded Wnt ligands and gradients of Fat-Dachsous pathway gene expression are attractive candidates for global planar polarity orientation cues. However, the role of Wnt gradients as tissue-level cues in regulating planar polarity in the developing *Drosophila* wing remains under debate: while a previous study demonstrated that Wnt expression can regulate global polarity orientation[14], two independent studies reported that removal of endogenous Wnts does not cause planar polarity defects[15,16]. The Fat-Dachsous pathway components Dachsous and Four-jointed are expressed in opposing

**Fig. 1 | Tissue-level Fz polarity, stability and stress evolution. a** *Drosophila* pupal wing blade morphogenesis begins at 16 hAPF with tissue tension arising from wing-hinge contraction, elongating wing blade along the PD-axis. Tension peaks at 24 hAPF, diminishing by 27 hAPF. Concurrently, core planar proteins reorient globally from AP to PD polarity. Yellow arrows indicate hinge contraction; blue-red arrows show local polarity. Green/blue boxes mark proximal/distal regions analysed. **a'** Schematic of core planar polarity proteins asymmetry in wild-type (WT) pupal wing cells, leading to trichome formation at the distal junctions. **a''** Asymmetrical localisation of core planar polarity proteins at apical cell junctions. **b** Coarse-grain polarity pattern in the proximal region (green box, **a**) of WT wings expressing Fz::EGFP and Sqh::3xmKate at 24, 28 and 32 hAPF. Yellow bars represent the magnitude (length) and angle (orientation) of planar polarisation for a group of cells. **b'** Average polarity magnitudes for Fz::EGFP and Sqh::3xmKate at 24, 28 and 32 hAPF. **c** Static snapshots before and 120 s after circular ablation in the proximal region of WT wings at 24, 28, and 32 hAPF. Grey-dotted ROI outlines the ablation;

orange ellipse indicates elliptical deformation extent. Blue and red arrows represent major and minor axes of the ellipse. **c'** Temporal evolution of relative amount of stable Fz (fluorescent timer analysis) and anisotropic recoil velocity in WT wings over time. **d** Coarse-grain polarity pattern in the distal region (blue box, **a**) of WT wings expressing Fz::EGFP and Sqh::3xmKate at 24, 28 and 32 hAPF. **e** Average Sqh::3xmKate polarity magnitudes in proximal and distal regions over time. **P = 0.0042 and *P = 0.0112. **f** Average cell eccentricity across regions and time points. ***P = 0.0002. **g** Fz::EGFP polarity angle variance in proximal and distal regions at 32 hAPF. Dot indicates mean per wing. Unpaired *t*-test, two-tailed, **P = 0.0015. **h** Relative stable Fz amounts across regions and time points. **P = 0.0061, *P = 0.0083. The number of wings examined is indicated. One-way ANOVA test, comparing proximal and distal wings for each time point. ns, not significant. Error bars are SEM. Microscopy images: anterior bottom, proximal left unless stated. Scale bars 5 µm. Source data are provided in the Source Data file.

gradients across the developing wing and knockout of *fat* or *dachsous* activity results in severe planar polarity defects[17–19]. However, uniform Dachsous expression is sufficient to effectively establish correctly oriented planar polarity[20–23]. Moreover, cell morphogenetic events such as cell elongation and oriented cell division are severely perturbed in the *dachsous* mutant wing, suggesting that it influences planar polarity through its role in regulating tissue epithelial dynamics[24].

Anisotropic tissue stress generated during tissue morphogenesis has emerged as a compelling tissue-level cue capable of coordinating cell behaviour and influencing planar polarity orientation across multiple cell distances. During *Drosophila* pupal wing development, the contraction of the wing hinge, starting around 16 hAPF, generates anisotropic tissue stress along the proximal-distal (PD) axis, resulting in the elongation of the wing blade[24,25] (Fig. 1a). This tension continues to increase until it peaks at 24 hAPF and then gradually diminishes by 27 hAPF. At the same time, core planar polarity reorients its axis, from a predominantly anterior-posterior (AP)-oriented to a PD-oriented alignment (Fig. 1a). Moreover, altering anisotropic tissue stress is sufficient to perturb tissue dynamics and planar polarity alignment in *Drosophila*, mouse and *Xenopus*, suggesting a causal link[24,26,27].

One potential mechanism by which anisotropic tissue stress could influence polarity orientation involves polarised cell rearrangements, also known as T1 transitions. This process coincides with polarity reorientation in the *Drosophila* wing and polarity establishment in the developing mouse epidermis[24,26]. During oriented T1 transition events, a junction perpendicular to the tissue stress axis contracts while a new junction forms and elongates parallel to the tissue stress axis, as a mechanism to alleviate tissue stress[28,29]. Mechanistically, it is proposed that core protein complexes are preferentially maintained on existing junctions and slow to accumulate on newly formed junctions generated by T1 transitions, this then contributing to subsequent polarisation along the tissue stress axis[24,26]. Indeed, altering T1 transition orientation via wing-hinge severing results in polarity alignment disorder[24]. However, the reason behind the slow accumulation of core protein complexes at newly formed junctions after a T1 transition is not understood.

Additionally, anisotropic tissue stress might orient polarity by affecting the orientation of microtubules. Several studies have reported the role of polarised microtubules in the directional transport of core polarity proteins to distal cell junctions in the pupal wing[30–33]. The wing blade elongation that coincides with PD planar polarity reorientation results in cells becoming PD elongated[24] and adopting a PD-oriented microtubule alignment[30–33]. These polarised microtubules may facilitate the delivery of new proteins to distal (vertical) junctions. However, no mechanisms have been reported by which pre-existing core proteins might be removed from horizontal junctions that are oriented parallel to the PD-axis, leading to a reorientation of polarity. A possible explanation might involve core protein degradation on

horizontal junctions over time. However, both cell elongation and microtubule polarisation gradually decrease several hours before the peak of core protein polarisation. This suggests that alternative mechanisms are in place to sustain the established polarity, for example, by continuously preventing the accumulation of core proteins on horizontal junctions. These could involve self-organising cellular mechanisms whereby higher accumulation of proteins on vertical junctions inhibits accumulation on horizontal junctions[34]. Alternatively, other tissue stress-dependent mechanisms may play a role in this process.

In this study, we use the *Drosophila* pupal wing to elucidate the molecular mechanisms governing the effects of mechanical tissue stress on global core planar polarity alignment. We provide evidence that reduction in anisotropic tissue stress leads to the disruption of polarity alignment along the PD-axis, and concurrently increases the stability of the essential core protein complex component Fz. Furthermore, we observe that the rate of Fz accumulation on nascent junctions after T1 transitions is dependent on the level of tissue stress anisotropy: Fz accumulates more swiftly on new junctions under conditions of reduced tissue stress, and more slowly when stress is high. Detailed examination of tissue dynamics uncovered a gradient of cell flow, characterised by differential velocities across adjacent cell rows, which emerges alongside increasing tissue stress anisotropy during pupal wing development. This cell flow gradient decreases when anisotropic stress is reduced, leading to enhanced Fz stability. Conversely, acutely increasing the cell flow gradient reduces Fz stability on junctions aligned with the flow axis. This destabilisation is sufficient to reorient the planar polarity axis, making it parallel to the flow axis. In normal development, high gradients of cell flow, indicative of intensified relative movement among adjacent cell rows, appear to facilitate the destabilisation of core protein complexes at junctions aligned with the direction of cell movement (along the PD-axis) and therefore play a role in reinforcing Fz polarity along the PD-axis. We surmise that this process is driven by shear stress at intercellular junctions, induced by the relative movement of cells, leading to the dissociation and remodelling of transmembrane complexes. We propose that gradients of cell flow play a critical role in establishing and maintaining planar polarity alignment throughout the tissue.

## Results

### Tissue stress and Fz dynamics during wing morphogenesis

Initially, we investigated the dynamic changes in tissue stress and in core planar polarity protein stability and polarity, during the 24–32 hAPF period of pupal wing development, preceding initiation of wing hair (trichome) formation. Throughout this stage, cells undergo rearrangements that help define the final wing shape, without the confounding effects of cell division and apoptosis[25].

A polarised cellular distribution (anisotropy) of Myosin-II (Sqh in flies) is known to correlate with tissue stress in the *Drosophila* wing

epithelium[35–38]. Furthermore, Kale et al.[39] demonstrated a significant correlation between junctional Myosin-II density and junctional tension (as measured by recoil velocity laser ablation) and Myosin-II is enriched on junctions under tension due to tissue stretching leading to increased Myosin-II polarisation[40]. This indicates that Myosin-II can be used as a readout of the distribution of junctional tension. Similarly, we used Fz stability as a readout for stability of core planar polarity complexes[41,42].

We conducted time-lapse imaging on the proximal-posterior region (below longitudinal vein 5, green box in Fig. 1a) of wild-type wings expressing Fz::EGFP[42] and Sqh::3xmKate[43]. Between 24 and 28 hAPF, Myosin-II displayed significant polarised enrichment at cell junctions, with a higher level of Myosin-II on horizontal junctions (oriented parallel to the PD-axis) compared to vertical junctions (oriented parallel to the AP-axis) (Fig. 1b). This suggests that horizontal junctions experience greater tension than vertical junctions, as reported previously[37]. To examine the associated anisotropic tissue stress, we analysed coarse-grained Myosin-II polarity. From 24 to 28 hAPF, Myosin-II polarity predominantly aligns with the AP-axis, indicating that tissue stress is more intense along the PD-axis (yellow lines, Fig. 1b). To follow the evolution of tissue stress anisotropy over time, we quantified average Myosin-II polarity magnitude[40,44]. This decreased from 24 hAPF onwards, indicative of a relaxation in tissue stress anisotropy, which can be attributed to the gradual reduction of wing-hinge contraction starting from 20 hAPF onwards[24,37,45] (Fig. 1b, b'). Conversely, from 24 to 32 hAPF, analysis of coarse-grained Fz::EGFP polarity showed an increasingly coordinated alignment of Fz among groups of cells along the PD-axis (yellow lines Fig. 1b). In addition, Fz::EGFP exhibited an increasing asymmetric enrichment on vertical junctions over time, as evidenced by the increasing average Fz::EGFP polarity magnitude (Fig. 1b') in line with previous studies[24,41,44,46].

In the past, tissue stress anisotropy has been assessed using laser ablation techniques that rely on the measurement of the initial retraction velocity[25,35,36,47–49]. For additional assessment of the temporal progression of tissue stress anisotropy during the late stages of pupal wing morphogenesis, we performed a circular laser ablation on the proximal-posterior region of the pupal wing. The magnitude of stress anisotropy was determined by measuring the difference between the initial recoil velocity of the ablated circle measured along the semi-major and semi-minor axes respectively (Fig. 1c and Supplementary Fig. 1a). Our experimental results revealed a gradual relaxation of tissue stress anisotropy from 24 to 32 hAPF (Fig. 1c, c'), consistent with previous findings[25,45] and our Myosin-II polarity data (Fig. 1b').

In addition to Fz polarity, there is a gradual increase in Fz stability over the same developmental period during which tissue stress anisotropy decreases, implying a potential link between Fz destabilisation and tissue stress[46]. Therefore, we evaluated the stable proportion of Fz protein localised at cell junctions across time using a Fz tandem fluorescence timer fusion (Supplementary Fig. 1b, b' and "Methods"). Aligning with the previous results obtained using Fluorescence Recovery After Photobleaching (FRAP) analysis[46], our observations also show an increase in the relative stable Fz from 24 to 32 hAPF (Fig. 1c' and Supplementary Fig. 1f, g). Hence, in the study, we have adopted the fluorescence timer to quantitatively assess the relative concentration of stable Fz across cell junctions.

Taking these results together, we find a striking negative correlation between the evolution of anisotropy of tissue stress, and Fz polarity and stability between 24 and 32 hAPF in the proximal wing (Fig. 1b', c'). To strengthen this finding, we also looked in the distal wing region. Previous work demonstrated a difference in tissue stress anisotropy between different regions of the pupal wing[25]. Indeed, we found that Myosin-II polarity was significantly higher at 24 and 28 hAPF in the proximal region as compared to the distal region (Fig. 1d, e). Also, cells in the proximal regions were more elongated as compared to distal regions at 24 hAPF (Fig. 1f). By 32 hAPF, Myosin-II polarity and

cell eccentricity were not significantly different between the different regions (Fig. 1e, f). Comparing local Fz polarity coordination between different regions of the pupal wing, the proximal wing region also exhibited significantly higher local polarity coordination along the PD-axis (where higher polarity angle variance indicates higher variance in polarity angle of all cells within the image) as compared to the distal region at 32 hAPF (Fig. 1g and Supplementary Fig. 1c). Conversely, average Fz polarity magnitude was not significantly different between these wing regions (Supplementary Fig. 1d, e). Notably, we also observed a lower stable proportion of Fz in the proximal wing region compared to the distal region at 24 and 28 hAPF but not significantly different by 32 hAPF (Fig. 1h). Overall, our results reveal a correlation in the proximal wing between higher tissue stress anisotropy and less stable Fz at 24–28 hAPF, with higher Fz local polarity coordination by 32 hAPF. This is consistent with the idea that an important role of anisotropy of tissue stress is to orient core protein polarity relative to the tissue axes, with higher stress increasing local coordination.

## Tissue stress manipulation affects Fz polarity and stability

A potential explanation for the correlation between tissue stress anisotropy and Fz polarity coordination is that tissue stress anisotropy directly regulates Fz polarity and stability on cell junctions.

To investigate, we employed physical manipulation to acutely reduce tissue stress anisotropy by physically detaching the wing blade from the hinge region using forceps at 20 hAPF, as done in a previous study[24] (Fig. 2a). Severed wings displayed reduced anisotropic tissue stress, as indicated by a significant decrease in Myosin-II polarity at 28 hAPF in comparison to control wings (Fig. 2b). This led to a marked decrease in the cell apical area and eccentricity compared to control wings (Supplementary Fig. 2a–b')[24].

By 32 hAPF, tissue stress anisotropy in severed wings was not significantly different from wild-type wings (Fig. 2b). We examined Fz polarity alignment in severed wings with reduced stress anisotropy and compared it to wild-type wings at 32 hAPF, before wing hair formation. In the severed wings, polarity was less aligned with the PD-axis (with higher angle variances in polarity alignment), whereas the wild-type wings exhibited more consistent alignment along the PD-axis (Fig. 2a, c). Interestingly, severed wings, characterised by a lower tissue stress anisotropy, exhibited a significantly higher stable proportion of Fz when compared to intact, non-detached control wings (Fig. 2d).

In parallel, we employed genetic manipulation of tissue stress within the pupal wing and analysed the impact on Fz behaviour. We reduced levels of Dumpy (Dpy), an extracellular matrix protein that attaches the wing to its surrounding cuticle and acts as a counterforce to hinge contraction, throughout the pupal wing by expressing double-stranded RNA under control of the *nubbin-Gal4* (*nub-Gal4*) driver. Loss of Dpy is associated with diminished anisotropic mechanical stress in the wing, leading to a decrease in the apical area of wing cells[25,45,50]. Subsequently, we assessed cell eccentricity and tissue stress anisotropy in *dpy-RNAi* wings from 24 to 32 hAPF (Fig. 2e). In *dpy-RNAi* wings, pronounced epidermal indentations appeared in the proximal region from 24 to 32 hAPF, making it challenging for laser ablation and live imaging studies. Therefore, we focused our analyses on the distal region of *dpy-RNAi* wings, which remained flat during this period, and compared it to the corresponding region in wild-type wings. As expected, *dpy-RNAi* wings, in which cells are less elongated as compared to the same region of control wild-type wings at 24 hAPF, also demonstrated significantly lower tissue stress anisotropy, evident by lower Myosin-II polarity at 24 hAPF (Fig. 2e–h). By 28 hAPF onwards, cell eccentricity and tissue stress anisotropy in *dpy-RNAi* were not significantly different from wild-type wings (Fig. 2e–h).

We then compared Fz polarity alignment in the distal region of *dpy-RNAi* wings with reduced stress anisotropy with the same region of control wild-type wings prior to wing hair formation at 32 hAPF. At this stage, polarity alignment in the *dpy-RNAi* wings was less aligned along

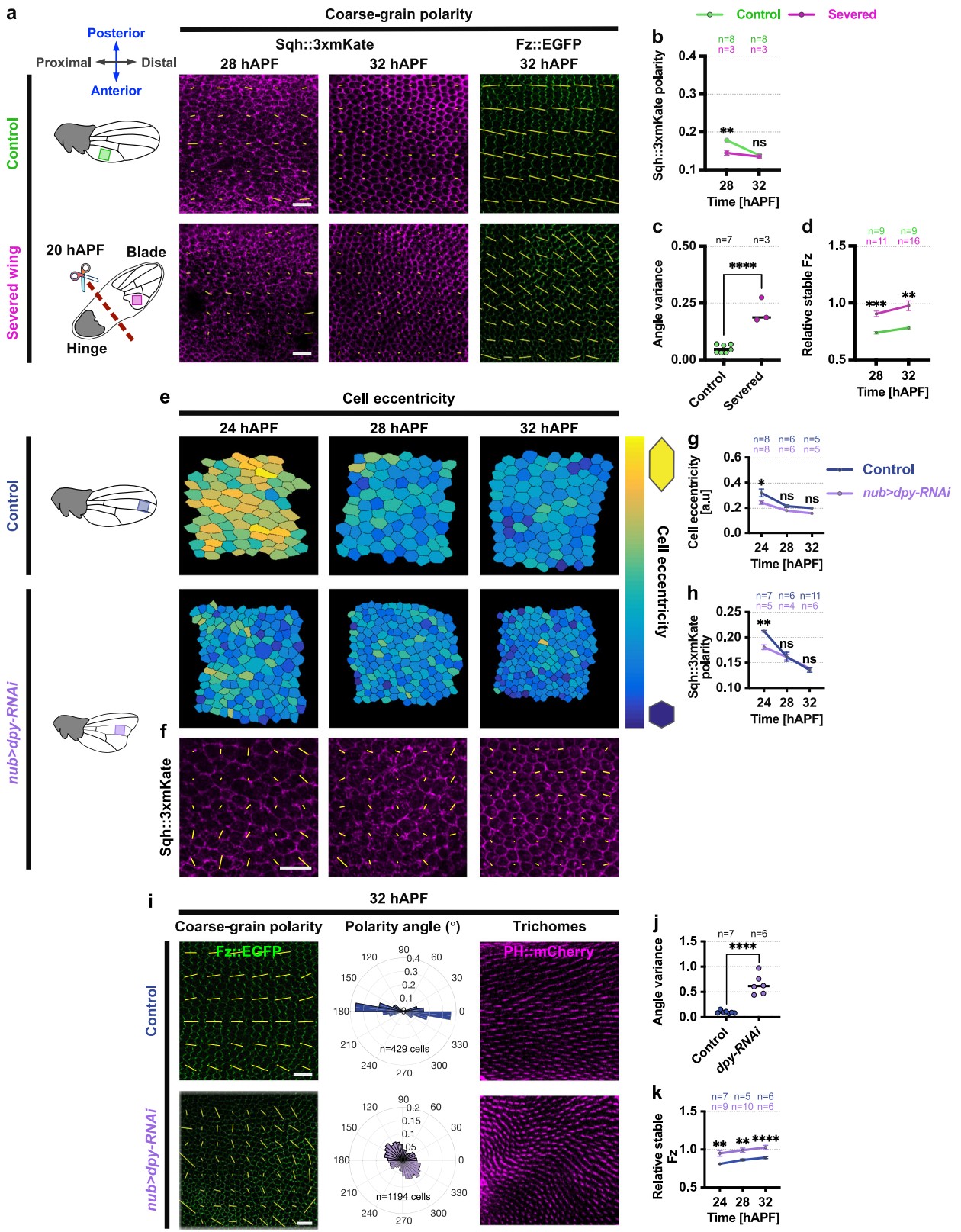

the PD-axis, while control wings displayed a more pronounced PD-oriented polarity alignment (Fig. 2i, j). We also confirmed that other regions in *dpy-RNAi* wings exhibit less pronounced PD-oriented polarity compared to corresponding regions in control wings (Supplementary Fig. 2c–d'). These results were further verified by examining the wing trichome orientation of *dpy-RNAi* and control wings using Pleckstrin homology domain marker tagged with mCherry

(PH::mCherry), with *dpy-RNAi* wings displaying a swirling trichome pattern (Fig. 2i). These outcomes collectively indicate that tissue stress anisotropy plays a pivotal role in establishing global polarity alignment, consistent with previous findings[24].

Additionally, we investigated Fz stability in control wild-type and *dpy-RNAi* wings of the same region from 24 to 32 hAPF. Interestingly, *dpy-RNAi* wings with lower tissue stress anisotropy led to a

**Fig. 2 | Manipulation of tissue stress affects Fz PD-polarity alignment and stability. a** Coarse-grain polarity in the proximal region of non-severed control (green box) and severed (pink box) wings expressing Sqh::3xmKate (28 and 32 hAPF) and Fz::EGFP (32 hAPF). Yellow bars indicate planar polarisation magnitude (length) and angle (orientation). **b** Quantification of average Sqh::3xmKate polarity magnitudes in control and severed wings at 28 and 32 hAPF. **P = 0.0092. **c** Fz::EGFP polarity angle variance in control and severed wings at 32 hAPF. Unpaired *t*-test, two-tailed ****P < 0.0001. **d** Stable Fz levels in control and severed wings at different time points. ***P = 0.0009, **P = 0.0024. **e** Distal regions of control (blue box) and *dpy-RNAi* wings (purple box) from 24 to 32 hAPF with cells colour-coded for shape eccentricity: yellow = highly eccentric, blue = circular. **f** Coarse-grain polarity in *dpy-RNAi* wings expressing Sqh::3xmKate (24–32 hAPF). Yellow bars represent planar polarisation magnitude and angle. **g** Cell eccentricity in distal regions of control and *dpy-RNAi* wings (24–32 hAPF). *P = 0.0292. **h** Average Sqh::3xmKate polarity

magnitudes in the distal region of control and *dpy-RNAi* wings (24–32 hAPF). **P = 0.0012. **i** Fz::EGFP coarse-grain polarity pattern in the distal region of control and *dpy-RNAi* wings shortly before wing hair formation (32 hAPF). Yellow bars depict planar polarisation; circular histograms show Fz::EGFP orientation. Actin-rich trichomes visualised with PH::mCherry. **j** Fz::EGFP polarity angle variance in the distal region of control and *dpy-RNAi* wings at 32 hAPF. Unpaired *t*-test, two-tailed ****P < 0.0001. **k** Stable Fz levels in distal regions of control and *dpy-RNAi* wings (24–32 hAPF). **P = 0.0089 (24 hAPF), **P = 0.0017 (28 hAPF), ****P < 0.0001 (32 hAPF). The number of wings (**b, c, d, g, h, j, k**) and cells (**i**) examined is indicated. Dot indicates mean per wing, and error bars are SEM. (**b, d**) One-way ANOVA test, comparing control and severed wings for each time point. (**g, h, k**) One-way ANOVA test, comparing control and *dpy-RNAi* wings for each time point. ns, not significant. Scale bars 10 μm. Source data are provided in the Source Data file.

pronounced increase in the stable proportion of Fz compared to control wild-type wings (Fig. 2k). Overall, these findings show that tissue with higher stress anisotropy exhibits more aligned polarity but also less stable Fz.

### Fz accumulates poorly on junctions under high tissue stress

The observed increase in PD-oriented Fz polarity and overall Fz stability over time suggests that Fz is becoming relatively more stable on vertical cell junctions as opposed to horizontal junctions, and that this is contributing to the increase in Fz polarity. As noted previously, the anisotropic tissue stress produced by hinge contraction causes polarised cell rearrangements or T1 transitions that result in production of new horizontal junctions and retention of existing old vertical junctions[24,26]. Hence, it has been suggested that retention of core planar polarity proteins on old junctions and slow accumulation on new junctions may be the mechanism by which hinge contraction promotes PD-oriented planar polarisation[24].

A prediction of this model that planar polarity proteins do not accumulate on new junctions, is that if a T1 transition results in a new vertical junction, then this should also exhibit no core protein accumulation. However, as the core pathway mediates feedback-dependent cell polarisation and cell-cell communication to locally coordinate planar polarity, an alternative expectation is that new vertical junctions will accumulate proteins if their neighbouring cells are already PD planar polarised, but again accumulation would be low on new horizontal junctions.

To investigate further, we first tracked newly formed vertical and horizontal junctions post-T1 transitions, pooling data occurring from 24 to 32 hAPF in the proximal-posterior region of wings. Subsequently, we monitored changes in Fz::EGFP fluorescence intensity on these nascent junctions over a span of -3 h (Fig. 3a). Fz exhibited faster accumulation on newly formed vertical junctions post-T1 transitions, while it was conspicuously slow to accumulate on newly formed horizontal junctions (Fig. 3a, c green lines). Furthermore, the elongation rate of both types of newly formed junctions following T1 transitions did not differ significantly (Fig. 3c'), consistent with the rate of Fz accumulation on these junctions not being influenced by changes in their length. Thus, these observations in the proximal region of the wing corroborate the previous finding[24] that core proteins accumulate poorly on newly formed horizontal junctions.

We then questioned whether anisotropy of tissue stress might affect the pattern of Fz accumulation. As the anisotropic tension within the tissue begins to diminish after 27 hAPF (Fig. 1c'), we divided our data on Fz accumulation rates on nascent junctions formed post-T1 transitions into two developmental stages: an early stage (24–28 hAPF) and a late stage (28–32 hAPF). Unexpectedly, this analysis showed that Fz gradually increased on nascent horizontal junctions at the late stage, albeit at a slower rate than on vertical junctions (Supplementary Fig. 3a, b). In fact, both nascent horizontal and vertical junctions formed during late stages showed a significantly higher Fz

accumulation rate than those formed in the earlier stage (Supplementary Fig. 3b). To examine the impact of Fz accumulation on its stability at cell junctions, we also quantified the stability of Fz across all horizontal and vertical junctions employing the fluorescence timer method. We found that at both 24 and 32 hAPF, vertical junctions, with higher rates of Fz accumulation, also exhibit a higher relative stable proportion of Fz as compared to horizontal junctions, where Fz accumulates at a slower rate (Supplementary Fig. 3c). Therefore, while the accumulation rate of Fz is consistently higher on new vertical junctions across all stages, we noted an increased rate of Fz accumulation on new horizontal junctions during the later stages (28–32 hAPF), when tissue stress is reduced.

Expanding our analysis to the distal region of wild-type wings, which experiences lower anisotropic tissue stress compared to the proximal region (Fig. 1e), we observed a similar gradual increase in Fz accumulation at horizontal junctions, although once more at a slower rate than at vertical junctions (Fig. 3b, c blue lines). To further investigate the influence of tissue stress anisotropy on Fz accumulation on nascent junctions formed post-T1 transitions, we examined Fz accumulation rates in wings with reduced PD-oriented tissue stress (*dpy-RNAi*). We observed that Fz accumulated more rapidly on both newly formed horizontal and vertical junctions in *dpy-RNAi* wings as compared to control wild-type wings (Fig. 3e, f). These results support the view that new junctions under high stress show lower rates of Fz accumulation.

Interestingly, we observed that in regions/genotypes characterised by lower tissue stress and increased Fz accumulation at newly forming horizontal junctions (i.e. the distal region of wild-type wings and *dpy-RNAi* wings), the relative stable amount of Fz between horizontal and vertical junctions did not show significant differences at 32 hAPF (Fig. 3d, g). This once again suggests that polarised tissue stress normally acts to reinforce Fz planar polarity by decreasing Fz stability on horizontal junctions.

Collectively, these results suggest a more complex scenario than a simplistic model where Fz does not accumulate at newly formed horizontal junctions. Rather, they fit a pattern whereby Fz accumulation is hindered at horizontal junctions under high anisotropic tissue stress. Conversely, Fz accumulates more rapidly at horizontal junctions in *dpy-RNAi* wings, where tissue stress anisotropy is minimal. In all situations, horizontal junctions with higher rates of Fz accumulation also exhibit a higher proportion of stable Fz. Overall, we conclude that high anisotropic tissue stress impedes the stability and accumulation of Fz on cell junctions.

### Anisotropic tissue stress induces a gradient of cell flow

To provide a mechanistic explanation for high tissue stress anisotropy hindering Fz accumulation on cell junctions, we first carried out a detailed analysis of the effects of tissue stress anisotropy on cell behaviours in different regions of the pupal wing. Previous studies shed light on the presence of active tissue flow during *Drosophila*

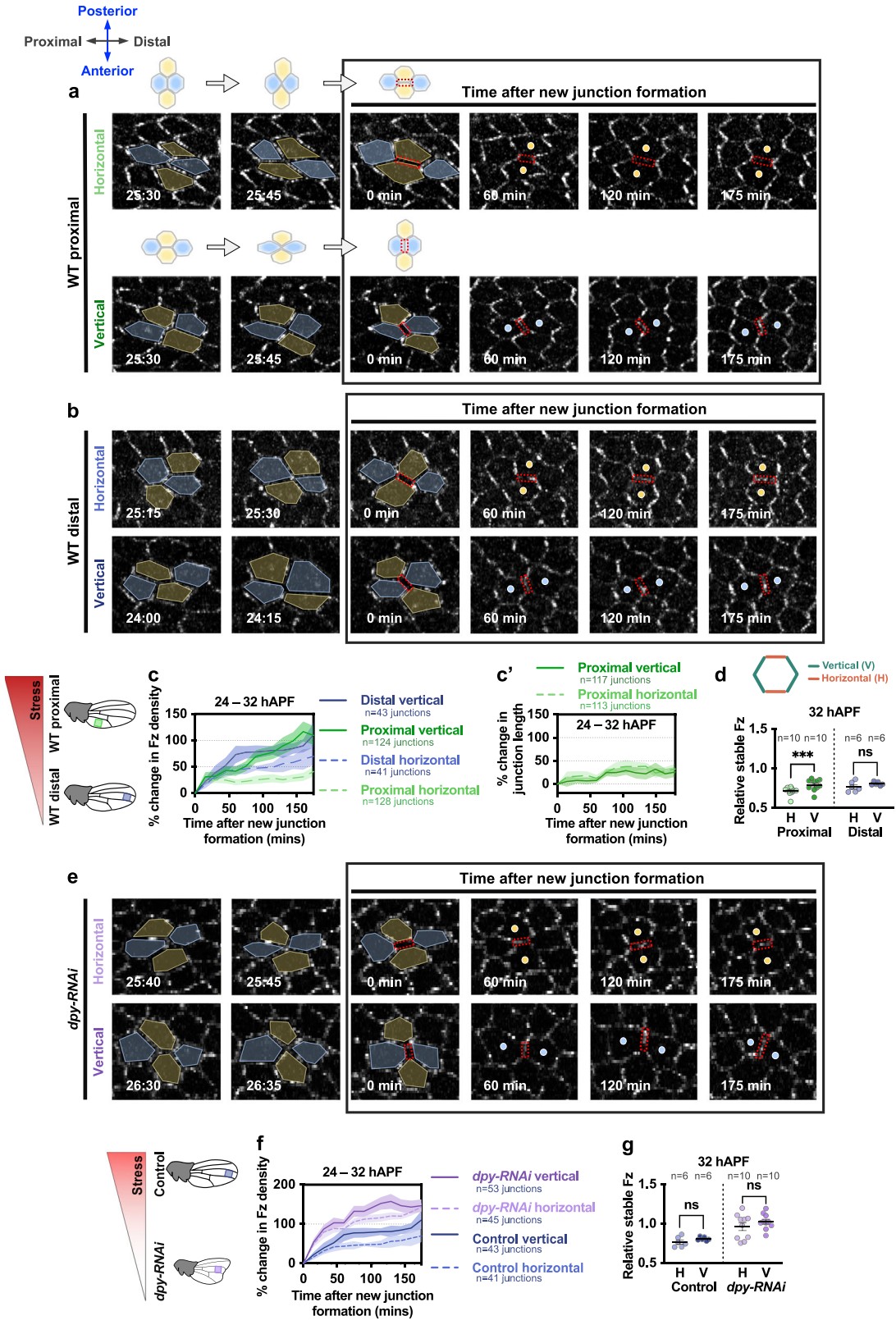

First, we analysed the proximal region of the wild-type pupal wing where Fz is slow to accumulate on nascent horizontal junctions. Here, we observed a unidirectional movement of cells flowing toward the proximal direction of the wing from 24 to 34 hAPF (Fig. 4a). We performed detailed analysis of the pattern of cell displacement in this subregion over time by tracking groups of cells for a 2-h period at 24, 28 and 32 hAPF, respectively. This revealed a steep velocity gradient

pupal wing morphogenesis, driven by anisotropic tissue stress stemming from the contraction of wing-hinge cells between 16 and 24 hAPF, leading to the wing blade narrowing along the AP-axis and elongating along the PD-axis[24,25,50]. In particular, different regions of the wing blade exhibit variability in the velocity of cell flow, with the fastest flow observed in the central region and the slowest in the distal region of the wing blade[24].

**Fig. 3 | Fz accumulation on newly formed junctions post-T1 transition.**
**a**, **b** Time-lapse imaging of Fz::EGFP in (**a**) proximal and (**b**) distal WT wing regions over a 3-h period, tracking the dynamics of Fz::EGFP on newly formed horizontal and vertical junctions (denoted by red boxes) shared between two adjacent cells (yellow/blue circles) following the T1 transition event. All tracked T1 transition events occurred between 24 and 32 hAPF. **c**, **c'** Graphs depicting (c) the percentage change in Fz density on newly formed junctions in the proximal (green box) and distal (blue box) regions of WT wings, and (c') the percentage change in junction length of all new junctions in the proximal WT region over a 3-hour period following a post-T1 transition. **d** Relative amount of stable Fz on all horizontal (oriented parallel to the PD-axis) and vertical junctions (oriented parallel to the AP-axis) for both proximal and distal regions of WT wings at 32 hAPF. Paired *t*-test, two-tailed ***$P = 0.0005$; ns, not significant. **e** Time-lapse imaging of Fz::EGFP in *dpy-RNAi* wing over a 3-h period during T1 transition. New horizontal/vertical junctions (red boxes), shared between two adjacent cells (yellow/blue circles) following the T1 transition event, are tracked over time. **f** Graph depicting the percentage change in Fz density on newly formed junctions during a 3-h period post-T1 transition in the distal region of control (blue box) and *dpy-RNAi* (purple box) wings. **g** Relative amount of stable Fz on horizontal and vertical junctions for distal region of control and *dpy-RNAi* wings at 32 hAPF. Paired *t*-test, two-tailed; ns, not significant. The number of junctions (**c**, **c'**, **f**) and wings (**d**, **g**) examined is indicated. Dot indicates mean per wing, error bars are SEM. Source data are provided in the Source Data file.

between neighbouring rows of cell along the AP-axis, with the anterior rows of cells (adjacent to 5th longitudinal vein cells) displaying faster flow than the posterior rows of cells (adjacent to wing margin cells) at 24 hAPF (Fig. 4a). Moreover, the steepness of the velocity gradient between different cell layers gradually decreases from 24 to 32 APF (Fig. 4a and Supplementary Movie 1). Theoretical and numerical simulation studies have shown that gradient flow patterns could be produced by active stress and shear via cell rearrangement[51,52]. However, in *Drosophila* pupal wing tissue, a gradient of cell flow may serve as a stress-dissipation mechanism to relax tissue under tension. This gradient could result from variations in the wing's geometrical and structural properties, such as the significantly stiffer wing margin[53]. Overall, our observations reveal that the orientation of epithelial cell flows aligns with the axis of higher anisotropic stress (PD-axis) and the gradient of cell flow is greater at earlier stages when the anisotropy of tissue stress is higher (Fig. 4a).

In the observed tissue flow pattern, adjacent rows of cells move relative to each other in a direction parallel to the axis of tissue stress, namely, the PD-axis (Fig. 4d). To quantify this relative cell movement, we computed the tissue velocity gradient, which captures the extent of movement between neighbouring rows of cells along the PD-axis, hereafter referred to as velocity$_x$ gradient (Fig. 4d). A greater magnitude of the velocity$_x$ gradient signifies increased relative movement between neighbouring rows of cells along the PD-axis. To assess the variability in velocity gradients experienced by individual cells, we performed a local velocity$_x$ gradient analysis (referred to as coarse-grain velocity gradient) (Supplementary Fig. 4a, b and see "Methods"). The heatmaps of coarse-grain velocity gradients further revealed local variability in the magnitude of the velocity$_x$ gradient experienced by cells across different regions (Supplementary Fig. 4d).

Next, we computed the global tissue velocity gradient by averaging the velocity gradients across clusters of cells. This approach allowed us to capture a tissue-wide measurement of the velocity gradient while preserving some of the spatial structure of the data. To track the temporal evolution of the tissue velocity gradients over time, we computed instantaneous tissue velocity$_x$ gradient at 15-min intervals between 24 and 26 hAPF in the proximal and distal regions of the wing. Intriguingly, the proximal region, characterised by higher tissue stress, displayed a significantly higher velocity$_x$ gradient compared to the distal region, which exhibits lower tissue stress anisotropy (Fig. 4a, b, e and Supplementary Movie 2). We extended this analysis, measuring the average tissue velocity gradient for 2-h periods at later developmental stages. The proximal region exhibits a significantly higher velocity$_x$ gradient compared to the distal region at both 24 and 28 hAPF (Fig. 4a, b, f blue stars). However, by 32 hAPF, as tissue stress anisotropy diminished, both regions exhibited comparable tissue velocity$_x$ gradients (Fig. 4f blue stars). These findings show that, in the proximal region and during early developmental stages, where tissue stress anisotropy is more pronounced, there is greater relative movement between neighbouring rows of cells. In contrast, in the distal region and during later developmental stages, where tissue stress anisotropy is less pronounced, there is reduced relative movement of cells.

To further elucidate the causal relationship between tissue stress anisotropy and tissue flow patterns, we analysed the patterns of tissue flow in the distal region of *dpy-RNAi* wings that exhibit significantly lower tissue stress anisotropy as compared to both proximal and distal wild-type wings (Fig. 2h and Supplementary Fig. 4e, f). In *dpy-RNAi* wings, epithelial cells flowed proximally with greater velocity as compared to control distal wings at 24 hAPF (Fig. 4b, c and Supplementary Movie 3). The absence of counterforces from the Dpy-mediated wing blade-to-cuticle connection caused a swift retraction of the entire wing blade toward the wing hinge upon hinge contraction[25,50]. Despite the increased tissue flow velocity in *dpy-RNAi* wings, the instantaneous and average tissue velocity$_x$ gradients between neighbouring rows of cells in *dpy-RNAi* wings are significantly lower as compared to the proximal (24 and 28 hAPF) and distal regions (24 hAPF) of wild-type wings (Fig. 4e, f purple and grey stars). As tissue stress anisotropy diminished by 32 hAPF, both control wild-type and *dpy-RNAi* wings exhibited comparable tissue velocity$_x$ gradients (Fig. 4f). Notably, the tissue shear rate obtained using the triangulation method[25] gave similar results to our tissue velocity gradient measurements across all genotypes at 24 hAPF (compare Fig. 4f and Supplementary Fig. 4c).

Additionally, we further examined the pattern of tissue flow in severed wings with reduced tissue stress anisotropy. Evaluating the gradient of cell flow velocity in severed wings revealed a retrograde distal flow direction (Fig. 4g and Supplementary Movie 4), due to the proximally-detached wing blade undergoing retraction towards its distal end. Similar to *dpy-RNAi* wings, the proximal region of severed wings exhibits a significantly lower velocity$_x$ gradient as compared to that of the same region of wild-type control wings at 28 hAPF (Fig. 4h). However, as tissue stress anisotropy diminished by 32 hAPF (Fig. 2b), both severed and control wings displayed comparable tissue velocity$_x$ gradients (Fig. 4h). Taken together, these findings suggest an essential role of tissue stress anisotropy in shaping the tissue velocity gradient profile. Similar to *dpy-RNAi* wings, the relative stable amount of Fz between horizontal and vertical junctions in severed wings did not show significant differences at 32 hAPF (Fig. 4i), suggesting reduced tissue velocity gradient leads to loss of PD-polarity coordination in severed wings (Fig. 2a, c).

We also calculated the tissue velocity gradient (velocity$_y$ gradient) to capture the relative movement between neighbouring columns of cells along the AP-axis direction in different conditions (Supplementary Fig. 5a–e). Tissue velocity$_y$ gradients were not significantly different across the genotypes and time points, showing lack of correlation with Fz stability (Supplementary Fig. 5f–h). Moreover, it is worth noting that the magnitudes of the velocity$_y$ gradient in these tissues were relatively low compared to the velocity$_x$ gradient.

Overall, we observe a robust correlation between average tissue velocity$_x$ gradient and Fz stability across multiple experimental conditions, indicating that tissues with a lower velocity$_x$ gradient show higher Fz stability and vice versa (coefficient of determination, $R^2 = 0.7151$) (Fig. 4j). To examine how tissue velocity gradients locally affect Fz stability, we assessed changes in Fz stability across all horizontal junctions in proximal wild-type wings at the stage with highest gradient of cell flow (24–26 hAPF). First, we clustered the cells into

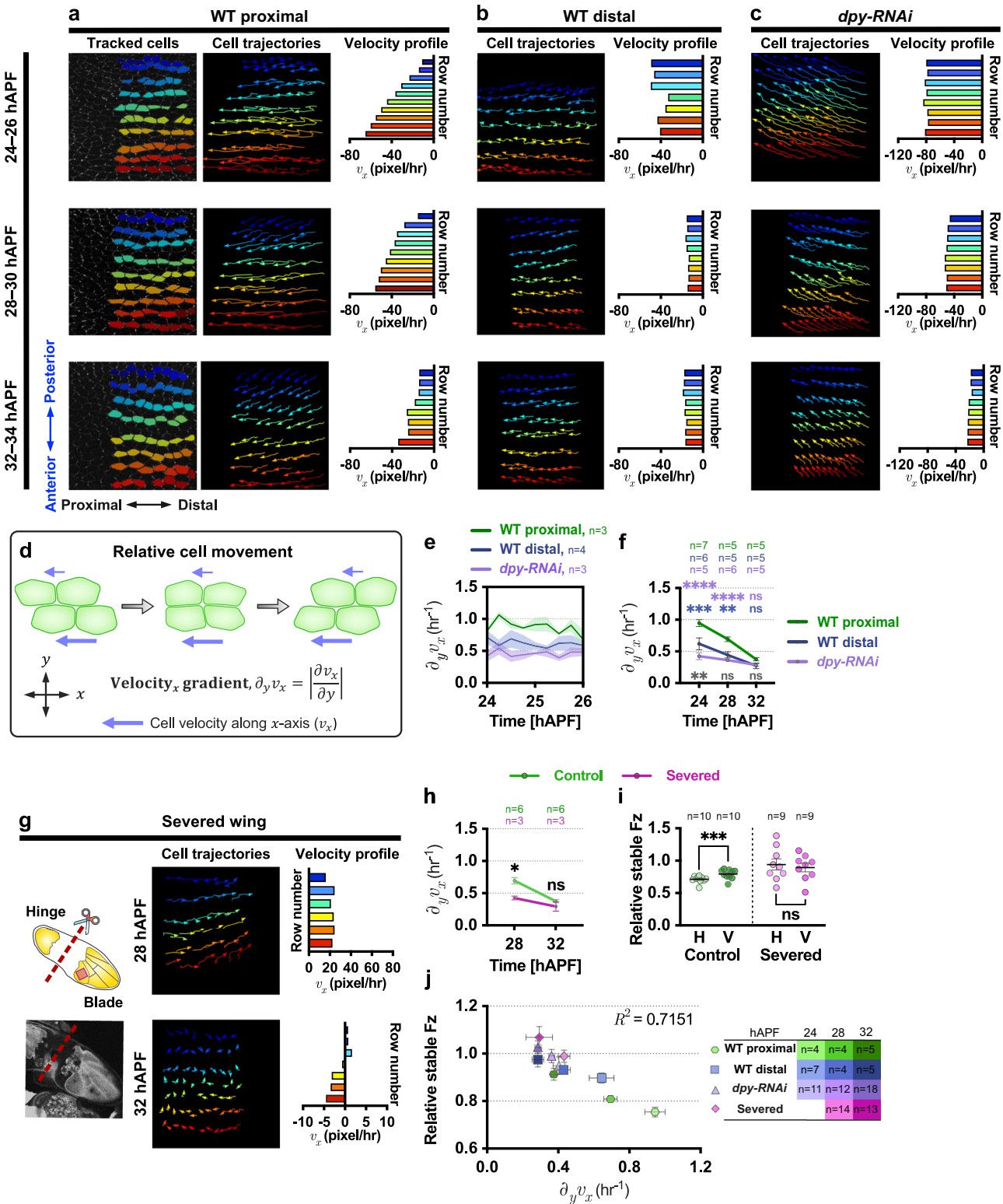

groups and computed the coarse-grain velocity gradient experienced by each cluster during this time period. We then determined the change in Fz stability within each cluster from 24 to 26 hAPF. Our analysis revealed that clusters of cells exposed to higher gradients of cell flow exhibited a greater reduction in Fz stability on horizontal junctions and vice versa, suggesting a correlation between the gradient of cell flow and Fz stability at junctions that are parallel to the flow direction (Supplementary Fig. 4g). In contrast, Fz stability on junctions perpendicular to the flow direction varied independently of the

magnitude of tissue velocity gradients in other regions (Supplementary Fig. 4g′). Collectively, these findings suggest that tissue velocity gradients affect the turnover of Fz, particularly on junctions that are parallel to the direction of flow.

**Acute tissue gradients affect Fz stability and polarity**
To examine the causal relationship between tissue velocity gradients and Fz turnover, we designed experiments to acutely induce a velocity gradient in the tissue. We did this on *dpy-RNAi* background wings at

**Fig. 4 | Anisotropic tissue stress influences epithelial flow patterns. a**, **b**, **c** Analysis of tissue flow patterns in (**a**) WT proximal and (**b**) WT distal regions and (**c**) distal *dpy-RNAi* wings (24–32 hAPF). Segmented and tracked cells at each time point were pseudo-coloured by rows (*y*-axis of the image). Cell trajectories were monitored over 90–120 min, with arrows indicating cell flow velocity direction. Velocity profile plots depict average velocity per row. **d** Relative cell movement, where adjacent rows of cells move relative to each other, visualised with blue arrows indicating magnitude (length) and direction (orientation) of velocity. Tissue velocity$_x$ gradient measures the partial derivative of the velocity component in the *x*-direction with respect to the *y*-direction. **e** Temporal evolution of instantaneous velocity$_x$ gradients (24–26 hAPF) for WT proximal, WT distal, and *dpy-RNAi* wings. **f** Average tissue velocity$_x$ gradient in WT proximal, WT distal and *dpy-RNAi* wings at indicated time points. Blue: WT distal vs WT proximal ***$P$ = 0.001, **$P$ = 0.0048; grey: WT distal vs *dpy-RNAi* **$P$ = 0.0082; purple: WT proximal vs d*py-RNAi* ****$P$ < 0.0001. **g** Cartoon (top) depicts wing-hinge detachment from the blade at

20 hAPF (red dotted line). Confocal image (bottom) shows a severed wing at 32 hAPF. Analysis of tissue flow patterns in severed wings (28–32 hAPF). Cell trajectories in imaged region (red box) were monitored for 2 h, with velocity profile plots displaying the average velocity for each row of cells. **h** Average tissue velocity$_x$ gradients in control and severed wings across time points. *$P$ = 0.0116. **i** Relative stable Fz amounts at horizontal and vertical junctions in control and severed wings at 32 hAPF. Dot indicates mean per wing. Paired *t*-test, two-tailed ***$P$ = 0.0005. **j** Correlation between tissue velocity$_x$ gradient and relative stable Fz across genotypes (in different colours and shapes) and developmental time points (in different colour gradients). Dot represents averaged values from all wings per time point. $R^2$ indicates the coefficient of determination. The number of wings examined is indicated. Error bars are SEM. (**f**) One-way ANOVA test, comparing between WT proximal, WT distal and *dpy-RNAi* wings for each time point. (**h**) One-way ANOVA test, comparing control and severed wings for each time point. ns, not significant. Source data are provided in the Source Data file.

28 hAPF, as these exhibit low tissue stress anisotropy and velocity gradients, making it easier to manipulate cell behaviour (Fig. 2h, Supplementary Fig. 4c, f). In the control regions (located between the 2nd and 3rd longitudinal veins, denoted as L2 in Supplementary Fig. 6a), different rows of cells displayed nearly uniform flow velocity along the PD-axis, as evident from their cell trajectories and velocity profile (Fig. 5a). To acutely induce a velocity gradient in the tissue with respect to the AP-axis, we ectopically induced tissue shearing using laser ablation in a region located in between the 3rd and 4th longitudinal veins of same wing (henceforth referred to as PD-shear region, denoted as L3 in Supplementary Fig. 6a). Note that there were no significant differences in terms of cell eccentricity and relative Fz stability between these different regions of unablated *dpy-RNAi* wings at 30 hAPF (Supplementary Fig. 6b, c). Tissue contraction, induced by two vertically-oriented laser ablated wounds within the tissue, initiated a phase of cell elongation. Although cell elongation gradually decreased, we observed continued fluctuations in the tissue velocity$_x$ gradient, with adjacent layers of cells sliding past each other along the PD-axis (Fig. 5b and Supplementary Fig. 6d). In contrast, control regions consistently exhibited significantly lower instantaneous velocity gradient and cell elongation within the same timeframe (Supplementary Fig. 6d').

By performing time-lapse imaging of the cells within the PD-shear region post-ablation, we observed a steeper velocity profile with differential flow velocity along the PD-axis with a significantly higher average tissue velocity$_x$ gradient compared to the control regions of the same wing (Fig. 5a–c and Supplementary Movie 5), confirming that this assay can artificially induce tissue shearing in a low-shear environment. There was no notable difference in tissue velocity$_y$ gradients with respect to the PD-axis (Supplementary Fig. 6e).

To comprehend the effect of the acutely increased tissue velocity gradient on Fz stability, we measured the relative stable amount of Fz between control and PD-shear regions of the same wing at 2 h post-ablation and observed a lower stable proportion of Fz within the PD-shear regions (Fig. 5d). Furthermore, this reduction in Fz stability is not a consequence of laser ablation itself, as the relative amount of stable Fz remains unchanged both pre- and 5 min post-ablation (Supplementary Fig. 6f). The elevated velocity$_x$ gradient in the PD-shear regions compared to the control regions suggests that the destabilisation of Fz in the PD-shear regions might be due to the increased in the relative cell movement along the PD-axis (Fig. 5c).

We next asked whether the effect of relative cell movement on Fz stability might be mediated through altered rates of T1 transitions. We tracked the frequency of T1 transitions in the PD-shear region and compared them to the non-ablated region. Acute tissue ablation led to fewer T1 transition events compared to the unablated control region (Supplementary Fig. 6g), despite the observed lower Fz stability (Fig. 5d). We also found that the distal wild-type wing region had fewer T1 transitions compared to the proximal region (Supplementary

Fig. 6g'), but in this case correlating with higher Fz stability (Fig. 1h). These findings suggest that Fz stability is not directly influenced by the rate of T1 transitions.

To further explore the impact of an acutely increased tissue velocity gradient on Fz stability across different junctions, we categorised the junctions as horizontal or vertical based on their orientation with respect to the PD-axis (see "Methods") and compared the relative amounts of stable Fz (Fig. 5e). In the control regions, horizontal junctions exhibited comparable relative amounts of stable Fz compared to vertical junctions. This could be due to the swirling polarity phenotype as observed in *dpy-RNAi* wings (Fig. 2i, j). Conversely, within the PD-shear regions, horizontal junctions displayed significantly lower relative stable Fz compared to vertical junctions (Fig. 5e). This suggests that the relative movement between adjacent rows of cells along the PD-axis contributes to destabilising Fz specifically on junctions parallel to the PD-axis.

We further examined whether reorienting the tissue flow direction from along the PD-axis to the AP-axis could affect Fz stability on different junctions. In control regions, different columns of cells displayed nearly uniform flow velocity along the AP-axis, as evident from their cell trajectories and velocity profile (Fig. 5f). To acutely induce tissue velocity gradient with respect to the PD-axis, we generated tissue shearing by laser ablating two horizontal wounds on *dpy-RNAi* wings at 28 hAPF (referred to as AP-shear region) (Fig. 5g). Through the constrictions of two horizontally laser ablated wounds within the tissue, adjacent columns of cells were induced to slide past each other along the AP-axis (Fig. 5g and Supplementary Movie 6). Overall, there was an increase in average tissue velocity$_y$ gradient in the AP-shear regions as compared to the control regions (Fig. 5h), without any notable differences in terms of velocity$_x$ gradient (Supplementary Fig. 6h). As anticipated, this resulted in a significantly lower relative amount of stable Fz on the AP-shear regions as compared to control regions at 2 h post-ablation (Fig. 5i). With the increase in relative movement between adjacent columns of cells in the AP-shear regions, we asked whether this would lead to detectable destabilisation of Fz specifically on vertical junctions (parallel to the AP-axis). However, Fz stability was not significantly different between horizontal and vertical junctions in the control and AP-shear regions. We conclude that experimentally induced AP-shear results in Fz destabilisation, albeit the relatively weak induced tissue velocity$_y$ gradient is not sufficient to result in a detectable change between horizontal and vertical junctions.

One of the limitations of the experimental setup above is that *dpy-RNAi* wings exhibit a swirling Fz polarity phenotype at late developmental stages, making it harder to observe and compare changes in Fz stability on different junctions upon induction of a tissue velocity gradient. Hence, we wished to devise an experiment to acutely induce a tissue velocity gradient at an earlier pupal developmental stage i.e. around 16 hAPF. This is because at 16 hAPF, Fz polarity in *dpy-RNAi*

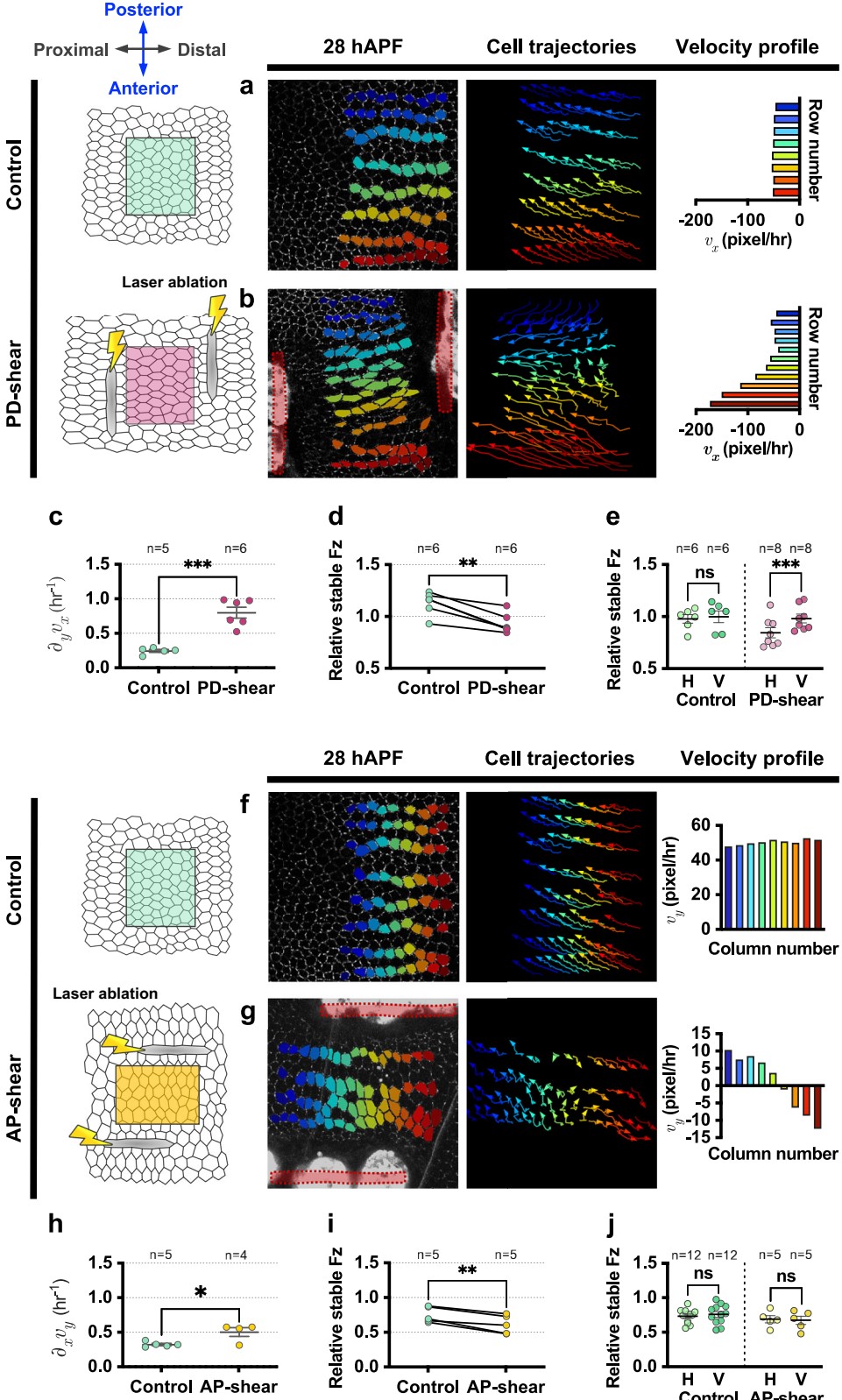

wings is predominantly oriented towards the wing margin, with an average polarity angle of 78° along the AP-axis (Fig. 6a). This AP-oriented Fz polarity indicates higher enrichment of Fz on horizontal junctions as compared to vertical junctions. Unexpectedly, analysing time-lapse images of the proximal-posterior region of *dpy-RNAi* wings (Control region), we observed that cells are undergoing rotational flow at 16−18 hAPF (Fig. 6b and Supplementary Movie 7).

We performed laser ablation to induce tissue shearing along the PD-axis in *dpy-RNAi* wings, leading to the loss of rotational flow and resulting in a pronounced tissue velocity_x gradient with respect to the AP-axis (PD-shear) from 16 to 18 hAPF (Fig. 6c, d and Supplementary Movie 7). Consequently, the stable proportion of Fz was reduced at 2 h post-ablation (18 hAPF) (Fig. 6e). In the control regions, horizontal junctions exhibited higher relative stable Fz than vertical junctions at

**Fig. 5 | Effects of acutely induced tissue velocity gradients on Fz stability. a, b** Analysis of tissue flow patterns in (**a**) unablated region of *dpy-RNAi* wing (Control) at 28 hAPF and (**b**) ablated region of *dpy-RNAi* wing (PD-shear) immediately post-ablation. Example of segmented and tracked cells at 28 hAPF (for control) or immediately after tissue shearing (for PD-region). Each row (*y*-axis of the image) of cells was tracked and pseudo-coloured based on its respective row. Cell trajectories were monitored for a 2-h period, with arrows indicating the direction of cell flow velocity. Velocity profile plots illustrate the average velocity of each row of cells. **c** Quantification of average tissue velocity$_x$ gradient for both control and PD-shear regions. Unpaired *t*-test, two-tailed ***$P = 0.0002$. **d** Relative stable amount of Fz in control and PD-shear regions 2-h post-ablation (-30 hAPF). Paired *t*-test, **$P = 0.0030$. **e** Relative amount of stable Fz on horizontal and vertical junctions for control and PD-shear regions 2-h post-ablation (-30 hAPF). Paired *t*-test, two-tailed ***$P = 0.0006$; ns, not significant. **f, g** Analysis of tissue flow patterns in (**f**) unablated region of *dpy-RNAi* wing (Control) at 28 hAPF and (**g**) ablated region of *dpy-RNAi* wing (AP-shear) immediately post-ablation. Example of segmented and tracked cells at the beginning of each developmental time points. Each column (*x*-axis of the image) of cells was tracked and pseudo-coloured based on its respective column. Cell trajectories were monitored over a 2-h period, with arrows indicating the direction of cell flow velocity. Velocity profile plots depict the average velocity of each column of cells. **h** Average tissue velocity$_y$ gradient for control and AP-shear regions. Unpaired *t*-test, two-tailed *$P = 0.0185$. **i** Relative stable amount of Fz in control and AP-shear regions 2-h post-ablation (-30 hAPF). Paired *t*-test, two-tailed **$P = 0.0048$. **j** Relative amount of stable Fz on horizontal and vertical junctions for control and AP-shear regions 2-h post-ablation (-30 hAPF). Paired *t*-test, two-tailed; ns, not significant. The number of wings examined is indicated. Dot indicates mean per wing, error bars are SEM. Source data are provided in the Source Data file.

18 hAPF, consistent with the AP-oriented Fz polarity with an average polarity angle of 70° (Fig. 6f, g). Notably, a reversal in Fz stability occurred in the PD-shear regions, where vertical junctions exhibited a higher relative stable Fz than horizontal junctions (Fig. 6f, f'). Accordingly, we observed a reorientation of Fz polarity angle from AP- to PD-orientation in the PD-shear regions, with an average polarity angle of 6° (Fig. 6h). This observation suggests that the tissue velocity gradient plays a role in influencing the orientation of Fz polarity. Overall, these findings support the contention that a tissue velocity gradient can destabilise Fz on junctions parallel to the axis of relative cell movement and thereby determine the global Fz polarity alignment across a tissue.

### Relative cell movement reduces Fz density and stability

In tissues with a high-velocity gradient, as rows of cells move relative to one another, horizontal junctions both shorten ('Junction shortening') and extend ('Junction extension') (Fig. 7a). Unlike typical T1 transitions, both shortening and extending junctions exhibit a similar orientation, which is parallel to the axis of tissue flow. This cyclical shortening and extension of horizontal junctions driven by the relative movement of cells (Fig. 7a, b), could be the cause of the reduced Fz localisation on horizontal junctions that leads to the establishment of PD-oriented polarity.

To investigate, we monitored changes in Fz::EGFP density on horizontal junctions that are undergoing relative cell movements between 24 and 28 hAPF. During the junction shortening phase, we observed a rapid decrease in Fz density correlating with the complete shrinkage of horizontal junctions (Fig. 7b, b', blue lines). Subsequently, junctions of the same orientation formed and extended over time (Fig. 7b). We then monitored Fz accumulation on extending horizontal junctions over a period of 120 min, revealing a consistently lower concentration of Fz with respect to other persisting junctions in the same cell (Fig. 7b', c). Taken together, these findings illustrate two key points: firstly, horizontal junctions that shorten and disappear during relative cell movement show a rapid decrease in Fz density; and secondly, Fz is slow to accumulate on extending horizontal junctions.

We speculated that changes in Fz density on shortening and extending horizontal junctions in tissues with a high-velocity gradient might be caused by the 'sliding' between neighbouring layers of cells. This would result in shear stress being transmitted across the horizontal junctions. Such shear stress might dissociate the transmembrane core protein complexes, similar to the detachment of other surface-bound macromolecules due to tangential forces[54]. Consequently, this would decrease the overall stability of Fz on horizontal junctions, contributing to the polarisation of Fz along the PD-axis. A recent study has reported that junctional Myosin-II could exert shear forces during junctional shrinkage, leading to the destabilisation of E-Cadherin-dependent adhesion complexes, and subsequently resulting in a planar distribution of these complexes[39].

We tracked sliding junctions shared between two neighbouring rows of cells moving relative to each other ('sliding junctions') between 24 and 28 hAPF. These sliding junctions were identified as those that underwent a transition of one or both vertices from 4-way to 3-way (Fig. 7d). To assess changes in Fz stability, we measured the stability before and after the transition, within an approximate 15-min interval. Due to local variability in the velocity gradient experienced by different cells, some horizontal junctions did not undergo sliding events, resulting in a low-velocity gradient at specific time points (Supplementary Fig. 4d and Supplementary Movie 8). We quantified Fz stability on horizontal junctions that did not undergo any transition, as a comparative control ('control junctions'). We detected a decrease in average Fz stability (by 24%) on the sliding junctions undergoing relative cell movement (Fig. 7e). Consistent with this, during instances of sliding, we occasionally observed that Fz puncta disappeared on these sliding junctions as cells moved past each other (Fig. 7d). Conversely, we saw an increase in Fz stability (by 10%) on the control junctions (Fig. 7e). Notably, most of the sliding junctions also exhibited a reduction in junction length (Fig. 7f). Yet, the change in Fz stability does not correlate with the change in junction length, as shrinking control junctions exhibited on average a higher amount of stable Fz (Fig. 7g). Interestingly, the reduction in Fz stability is caused by reduced levels of slow-maturing Fz::mKate, while levels of fast-maturing Fz::sfGFP remained largely unaffected (Fig. 7h). In contrast, Fz::mKate density mildly increased on control junctions (Fig. 7i). These findings suggest that relative cell movement plays a significant role in the dynamic turnover of long-lived populations of Fz which we speculate represent Fz stably bound into intercellular transmembrane complexes.

A key interaction partner of Fz in intercellular transmembrane complexes is Fmi (Fig. 1a"), which forms cadherin homodimers bridging neighbouring cells[55,56]. We therefore explored changes in Fmi density amid relative cell movement. Interestingly, Fmi density on sliding horizontal junctions decreases (by 15%) during relative cell movement while control junctions remained constant (Fig. 7j). This would explain consistently slow accumulation of Fmi on nascent horizontal junctions formed post-T1 transition compared to nascent vertical junctions (Fig. 7k). In line with this idea, we further explored how relative cell movement affects adhesion proteins of the adherens junctions, such as E-Cadherin, which forms trans homodimers with E-Cadherin molecules on adjacent cells to support epithelial adhesion. Similar to our observations with Fmi, we noted a modest reduction in E-Cadherin density on sliding junctions compared to control junctions, with a decrease of 11% (Supplementary Fig. 7a). The change in E-Cadherin density does not appear to correlate with the change in junction length, as shrinking control junctions led to an increase in E-Cadherin density (Supplementary Fig. 7b, c). The loss of E-Cadherin from sliding junctions is consistent with E-Cadherin being predominantly polarised along the PD-axis from 24 to 28 hAPF

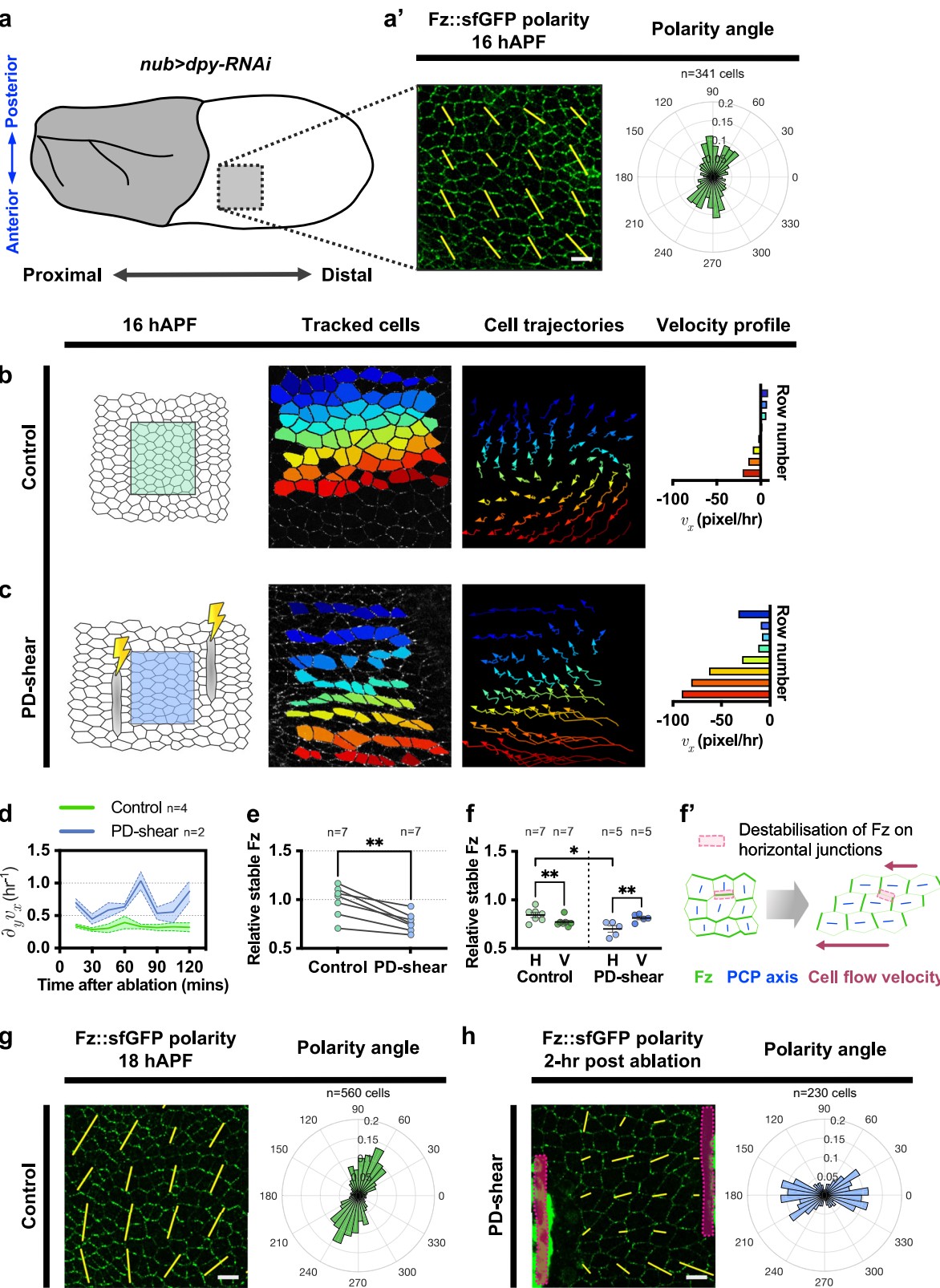

(Supplementary Fig. 7d). Combined together, these results suggest that relative cell movement results in overall reduction of core polarity protein intercellular transmembrane complexes (Fz and Fmi) and E-Cadherin on horizontal junctions, likely due to the influence of shear forces on these junctions.

Previous studies have reported that E-Cadherin membrane expression regulates adhesion strength, where higher E-Cadherin

expression correlates with stronger adhesion, and vice versa[57,58]. Additionally, Iyer et al.[45] demonstrated that wings with increased adhesion complex turnover, such as in *p120ctn* mutant wings, undergo faster cell shape changes, likely due to weakened cell-cell adhesion. Thus, it is plausible that altering cell adhesion strength by modulating E-Cadherin dynamics could influence Fz stability and polarity by affecting tissue flow during processes like cell sliding. For example,

**Fig. 6 | Effects of acutely induced tissue velocity gradients at 16 hAPF on Fz stability and polarity. a** Illustration of analysed control region (proximal-posterior) of *nub>dpy-RNAi* pupal wings at 16 hAPF (grey box). **a'** (Left) Fz::sfGFP coarse-grain polarity (yellow bars) at 16 hAPF of *dpy-RNAi* wing overlaid on confocal image. Yellow bars represent the magnitude (length) and angle (orientation) of planar polarisation for a group of cells. (Right) Circular weighted histogram plot displays the orientation of Fz::sfGFP polarity. **b, c** Analysis of tissue flow patterns in (**b**) unablated (Control) region at 16 hAPF and (**c**) ablated tissue shearing (PD-shear) region immediately post-ablation. Segmented and tracked cells were pseudo-coloured by rows (*y*-axis of the image). Cell trajectories were monitored over 90–120 min, with arrows indicating cell flow velocity direction. Velocity profile plots depict average velocity per row. **d** Temporal evolution of instantaneous velocity$_x$ gradient from 16 to 18 hAPF in control and PD-shear wing regions. **e** Relative stable amount of Fz in control and PD-shear regions at 18 hAPF (2 h post-ablation). Paired *t*-test, two-tailed **$P = 0.0017$. **f** Quantification of relative amount of stable Fz on horizontal and vertical junctions for both control and PD-shear regions 2-h post-ablation (~18 hAPF). Unpaired *t*-test, two-tailed *$P = 0.0354$ (H junctions control vs PD-shear). Paired *t*-test, two-tailed **$P = 0.002$ (Control H vs V junctions), **$P = 0.0097$ (PD-shear H vs V junctions). **f'** Illustration depicting destabilisation of Fz (shown in green) from horizontal junctions (pink box) during the movement of adjacent cells. Blue bars illustrate the Fz polarity orientation, while magenta arrows show the cell flow velocity. **g** (Left) Coarse-grain polarity pattern of control region expressing Fz::sfGFP at 18 hAPF. (Right) Circular weighted histogram plot displays the orientation of Fz::sfGFP polarity. **h** (Left) Coarse-grain polarity pattern of PD-shear region expressing Fz::sfGFP 2-h post-ablation. (Right) Circular weighted histogram plot displays the orientation of Fz::sfGFP polarity. For the analysis of polarity angle, both control and PD-shear regions are from the same wing region and developmental stage, and therefore should exhibit comparable Fz polarity alignment. The number of cells (**a', g, h**) and wings (**d, e, f**) examined is indicated. Dot indicates mean per wing, error bars are SEM. Scale bars 10 μm. Source data are provided in the Source Data file.

stronger adhesion might restrict cell sliding, promoting Fz stabilisation, whereas weaker adhesion could enhance cell movement and potentially destabilise Fz. Alternatively, cell adhesion strength may directly affect Fz stability, for instance by stabilising cell-cell contacts independent of tissue flow.

To test these hypotheses, we first examined the effect on tissue flow patterns and Fz stability of increasing E-Cadherin turnover and hence decreasing adhesion strength. Interestingly, we found that *p120ctn* mutant wings exhibited significantly lower tissue velocity gradients compared to wild-type wings from 24 to 34 hAPF (Fig. 8a, b and Supplementary Movie 9). We then measured Fz stability in *p120ctn* mutant wings, which exhibit lower E-Cadherin stability, and compared to control wings in the same proximal region at 28 hAPF. Using FRAP, we unexpectedly found that Fz::EGFP showed significantly greater stability in *p120ctn* mutant wings compared to wild-type control wings (Fig. 8c). Thus, contrary to our initial idea, this suggests that reduced cell adhesion enhances Fz stability rather than destabilising Fz. The *p120ctn* mutant wings also exhibited significantly lower local polarity coordination along the PD-axis (with higher polarity angle variance) as compared to the control wild-type wings at 32 hAPF (Fig. 8d, d'). We surmise that reduced adhesion may weaken the transmission of tissue stress, resulting in lower tissue flow gradients and consequently, increased Fz stability.

Given the putative role of tissue velocity gradients in regulating Fz stability, it is important to assess the effect of cell adhesion strength on Fz stability without the confounding influence of tissue stress anisotropy and cell flow gradients. To address this, we used wing-hinge severing to reduce tissue stress anisotropy and gradients of cell flow. We carried out the experiment both in wings with normal E-Cadherin levels and turnover (stronger adhesion) and in *p120ctn* mutant wings with higher E-cadherin turnover (weaker adhesion). Severed wild-type control wings demonstrated higher Fz::EGFP stability compared to non-severed wild-type controls (Fig. 8e), consistent with our previous findings (Fig. 2d). Interestingly, we found no significant difference in Fz::EGFP stability between severed control wings and severed *p120ctn* mutant wings (Fig. 8e), despite the expected differences in E-Cadherin turnover. This suggests that in the absence of cell flow gradients, E-Cadherin dynamics do not influence Fz stability.

## Discussion

In this study, we attempt to understand how mechanical cues align core planar polarisation across the PD-axis of the *Drosophila* wing epithelium. We propose that a gradient of cell flow, driven by anisotropic tissue stress, serves as a global cue directing overall planar polarisation by exerting shear forces on cell junctions aligned with the direction of flow. These forces destabilise core proteins at these junctions, thereby reinforcing the orientation of polarity (Fig. 8f). This flow and shear hypothesis is supported by several experimental

observations: (1) Fz stability is changed in response to manipulations of the gradient of cell flow; (2) an acute increase in tissue velocity gradient results in not only reduction in Fz stability on junctions aligned with the direction of flow but also Fz polarity reorientation; (3) as cells slide past one another, both the stability and density of junctional Fz are reduced.

First, we showed that manipulating tissue stress anisotropy results in the loss of PD-oriented polarity, in agreement with a previous report[24]. Despite the loss of PD-oriented Fz polarisation across the tissue, we note that the final average Fz polarity magnitude in tissues with different stress level anisotropies is not significantly different, suggesting that the ability of individual cells to polarise Fz is independent of tissue stress anisotropy. This is consistent with the proposed self-organising process of asymmetric cellular localisation driven by core protein-dependent feedback interactions[34].

Previous studies suggested that anisotropic tissue stress influences global planar polarisation by promoting oriented T1 transitions, which in turn lead to production of new cell junctions on which core proteins are slow to accumulate[24,26]. In the developing wing, this leads to a simple model whereby PD-oriented tissue stress induced by hinge contraction promotes PD-oriented T1 transitions and production of new horizontal junctions on which core protein levels are low. Critically examining this hypothesis, we found that on new junctions produced by T1 transitions, Fz accumulates slower on those that are parallel (horizontal) to the axis of tissue stress compared to those perpendicular (vertical) to it. However, we further observed that the rate at which Fz accumulates on new horizontal junctions post-T1 transitions is also influenced by the level of tissue stress anisotropy. Specifically, Fz can accumulate more rapidly on new horizontal junctions in tissue with lower stress anisotropy and more slowly in tissue with higher stress anisotropy. In tissues with reduced stress anisotropy, Fz accumulates rapidly on both nascent horizontal and vertical junctions post-T1-transition, resulting in polarity alignment disorder. Taken together, our data fail to support the hypothesis that global planar polarisation is strongly dependent on the orientation of T1 transitions. Instead, they point to a critical role for tissue stress anisotropy in hindering the accumulation of Fz on specific junctions.

In this regard, Aigouy et al.[24] described a role for anisotropy tissue stress in guiding tissue-level cell flows in the *Drosophila* pupal wing. Building on this, we explored how this stress-dependent mechanism might influence global planar polarisation. Intriguingly, we identified a novel phenomenon: a unidirectional gradient of cell flow along the AP-axis, where adjacent rows of cells move at different velocities. Indeed, manipulation of tissue stress anisotropy affects the gradient of cell flow, whereby stronger tissue stress anisotropy results in steeper gradient of cell flow. Strikingly, we also observed a correlation between the tissue velocity gradient and Fz stability. Furthermore, acutely increasing the tissue velocity gradient reduced Fz stability on junctions

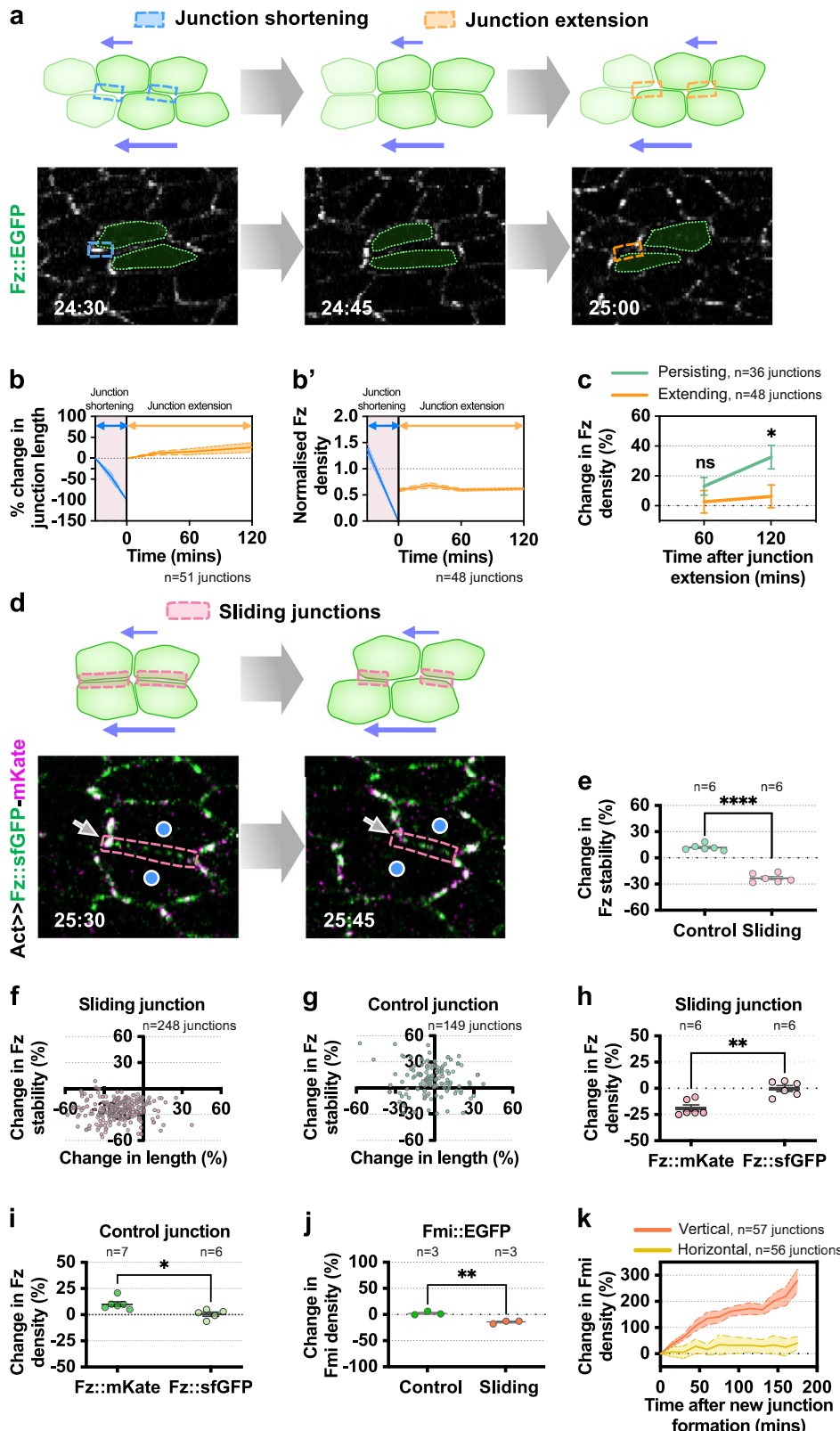

in the same orientation as cell flow. To establish sufficiency of this mechanism, we showed that at early developmental stages, an ectopic gradient of cell flow could reorient polarity.

With regard to our model that cell flows orient Fz polarity, it is interesting to note that even wild-type wing regions with a low gradient of cell flow can achieve normal polarity. For instance, the distal regions of wild-type wings, which exhibit a tissue flow gradient that is

only slightly higher than that of *dpy-RNAi* wings, exhibit predominantly PD-oriented Fz polarity, whereas *dpy-RNAi* wings display a severe swirl polarity phenotype. The most obvious explanation for this is that in the wild-type wing, higher cell flows in other regions of the wing help to set up PD-oriented polarity which is then transmitted to the distal wing via normal mechanisms of cell-cell propagation of polarity[9]. Furthermore, we found that the distal region of wild-type

**Fig. 7 | Relative cell movement leads to reduction of Fz and Fmi density and stability. a** (Top) Relative cell movement, where adjacent rows of cells move in relation to each other, resulting in junction shortening (blue box) and subsequent junction extension (orange box). (Bottom) Time-lapse of Fz::EGFP wings during relative cell movement event, tracking the Fz::EGFP dynamics on shortening and extending horizontal junctions. **b, b'** Quantification of (**b**) change in junction length (%) and (**b'**) normalised Fz density on shortening and extending junctions during relative cell movement (24–28 hAPF). **c** Quantification of change in Fz density (%) on persisting and extending junctions at 60 and 120 min post-junction extension. One-way ANOVA test, comparing persisting and extending junctions for each time point. *$P = 0.0251$; ns, not significant. **d** Illustration (top) of a junction shared (pink box) between two adjacent rows of cells (blue circles) during relative cell movement. Time-lapse imaging (bottom) of Fz::sfGFP-mKate wings during relative cell movement event, tracking the dynamics of Fz::sfGFP-mKate on sliding horizontal junctions. **e** Quantification of change in Fz stability (%) for both control and sliding junctions. All tracked relative cell movement events occurred between 24 and 28 hAPF. Unpaired *t*-test, two-tailed **** $P < 0.0001$. **f, g** Scatter plot showing the distribution of change in Fz stability (%) against change in (**f**) sliding junction length (%) and (**g**) control junction length (%). Each circle indicates an individual sliding junction. **h** Percentage of change in Fz::mKate and Fz::sfGFP densities on sliding junctions. Unpaired *t*-test, two-tailed **$P = 0.0010$. **i** Percentage of change in Fz::mKate and Fz::sfGFP densities on control junctions. Unpaired *t*-test, *$P = 0.0113$. **j** Percentage of change in Fmi density for control and sliding junctions. Unpaired *t*-test, two-tailed **$P = 0.003$. **k** Percentage of change in Fmi density on new horizontal and vertical junctions during a 3 h period post-T1 transition in control WT wings. The number of junctions (**b, b', c, f, g, k**) and wings (**e, h, i, j**) examined is indicated. (**e, h, i, j**) Dot indicates mean per wing. Error bars are SEM. Source data are provided in the Source Data file.

wings experiences higher velocity gradients from 20 to 22 hAPF, compared to *dpy-RNAi* wings (Supplementary Fig. 7e, e'). Consequently, these higher velocity gradients during earlier stages may contribute to the PD-oriented Fz polarity observed later in development.

We wished to understand the mechanisms underlying how relative cell movement caused by a gradient of cell flow affects Fz distribution and stability. Relative cell movement results in cells sliding relative to each other. We observed a decrease in Fz levels and stability on junctions undergoing such events. Friction between adjacent layers of cells during motion in a dense surrounding induces shear forces at the cell boundary which may influence the state of cell-cell adhesion contact[59]. Interestingly, it was the mature population of Fz protein that was reduced during relative movement of cells. Considering that Fz forms complexes with Fmi:Fmi dimers, and these complexes are more stable than Fz in its diffuse state[41], we speculate that the shear forces generated by relative cell movement directly disrupt transmembrane molecules, such as Fmi:Fmi intercellular dimers. This disruption would then lead to the destabilisation of the entire complexes containing Fz, similar to E-Cadherin being destabilised from contracting junctions that are under high shear forces[39]. Consistent with this, we also noted a reduction of Fmi and E-Cadherin adhesion complex density during relative cell movement. We propose that this removal of Fz from horizontal junctions experiencing shear forces is crucial for reorienting Fz polarity from being predominantly AP-oriented to PD-oriented. This mechanism would then serve as a long-range polarity cue to ensure global polarisation across the entire epithelium.

Understanding how shear forces influence intercellular complexes is crucial due to their involvement in tissue development and morphogenesis[39,60,61]. For example, shear forces arising from the opposing migration of axial mesoderm progenitors and neuroectodermal cells are vital for the correct placement of anterior neural progenitors, playing a significant role in shaping the embryo's overall structure[60]. Although measuring shear forces during cell movement in vivo is technically very difficult, the asymmetric distribution of junctional tension suggests their presence[39,62]. A notable challenge is differentiating between the effects of tensile and shear forces on Fz destabilisation, particularly since tissue shearing assays might activate both types of forces. A recent study showed that adhesion complexes such as E-Cadherin respond differently to tensile and shear forces[39]. Over long timescales, such as minutes to hours, under stress conditions cells subjected to high tensile forces may undergo intercalation or rearrangement as a stress dissipative mechanism[63,64]. Notably, in an acute tissue shearing assay, we observed that while cell elongation gradually decreases, the tissue velocity gradient continues to fluctuate over time. Consequently, we think that Fz destabilisation could be likely influenced more significantly by shear forces via relative cell movement than tensile forces. However, an impact of tensile forces on Fz destabilisation cannot be dismissed, and we plan to address this in future research.

In conclusion, we propose a new model to explain the role of mechanical stress anisotropy in establishing global planar polarisation, whereby shear forces induced via gradients of cell flow destabilise core proteins on junctions oriented parallel to the axis of cell flow. These insights open avenues for a deeper understanding of the role of shear forces and tissue-level cues in planar polarisation, advancing our understanding of long-range pattern formation during development.

## Methods

### Fly stocks and crosses

All fly strains used in this study are described in Table 1 and raised at 25 °C, unless otherwise stated. Flies were not selected based on sex, as no differences in the physical or molecular mechanisms of planar polarity in the pupal wing have been identified between males and females.

We previously generated a fusion construct containing Fz fused to slow-maturing mKate and fast-maturing sfGFP fluorescent proteins, under the control of the *ActinSC* promoter[65]. *fz::sfGFP-mKate* was expressed in a *fz^{P21}* mutant background after excision of the *FRT-STOP-FRT* cassette from *Act > STOP > fz::sfGFP-mKate* using hs-FLP following 2 h of heat shock at 37 °C at 0 hAPF. Previous work from our lab confirmed that pupal development halts for the period of heat shock based on the time of trichome emergence[46]. Consequently, the developmental time following heat shock was adjusted accordingly.

For all experiments except the temporal induction experiments, white prepupae were collected, rinsed in water and aged at 25 °C to the desired age before dissection and live imaging. Pupae at different developmental stages were used, as described in the text and figure legends.

### Method details

**Dissection and mounting of pupal wings for in vivo live imaging.** Following the protocol described by Classen et al.[66] pupae were affixed, dorsal side facing upwards, onto double-sided tape. The removal of the puparium case above a developing pupa was then performed using fine scissors and tweezers, ensuring the wing was exposed without causing harm to the pupa. The exposed pupal wing was coated with a drop of Halocarbon 700 oil. Finally, the exposed pupal wing was securely placed on a 2.5 cm diameter glass-bottomed dish (Iwaki), arranging the wing to face the coverslip.

**Live imaging of pupal wings.** Live imaging was conducted on an inverted Nikon A1 or Zeiss LSM880 confocal microscope.

The Nikon A1 was equipped with a 60× apochromatic oil objective lens (NA = 1.4) and GaAsP detectors. The microscope's pinhole was adjusted to 1.2 Airy Units (AU), and the stage temperature was consistently maintained at 25 °C. To ensure consistency across all imaging sessions, the laser power was routinely monitored and adjusted as required. The green and red fluorescence emissions were captured using a 488 nm laser combined with a 525–550 band-pass filter for

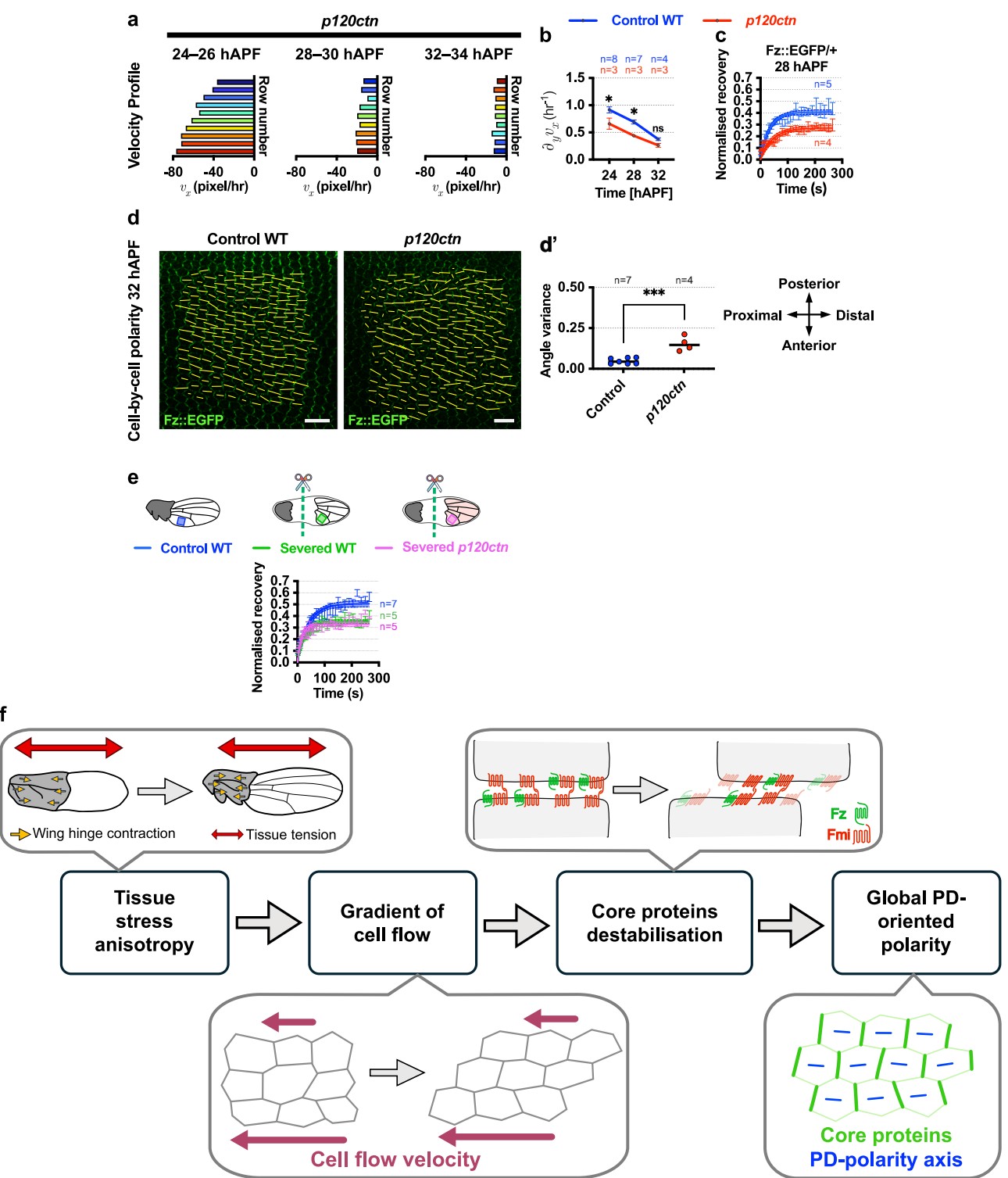

**Fig. 8 | Manipulating E-Cadherin turnover does not affect Fz stability. a** Analysis of tissue flow patterns in the proximal region of *p120ctn* wings from 24 to 32 hAPF. Velocity profile plots illustrate the average velocity of each row of cells. **b** Average tissue velocity$_x$ gradient of the proximal region of control WT and *p120ctn* wings from 24 to 32 hAPF. One-way ANOVA test, comparing control WT wings and *p120ctn* wings for each time point. *$P$ = 0.0208 (24 hAPF), *$P$ = 0.0180 (28 hAPF); ns, not significant. **c** Quantification of recovery fraction (normalised intensity) of control WT and *p120ctn* wings at 28 hAPF post photobleaching using Fluorescence Recovery After Photobleaching (FRAP) assay with $P$ < 0.0001. Solid lines indicate one-phase exponential fits. **d** Fz::EGFP cell-by-cell polarity nematics (represented by yellow bars) of control WT and *p120ctn* wings at 32 hAPF. Yellow bars represent the magnitude (length) and angle (orientation) of planar polarisation for each cell. **d'** Fz::EGFP polarity angle variance at 32 hAPF in control WT and *p120ctn* wings. Unpaired *t*-test, two-tailed ***$P$ = 0.0003. **e** Quantification of recovery fraction of control WT (blue box), severed WT (green box) and severed *p120ctn* (magenta box) wings at 24 hAPF post FRAP assay with $P$ < 0.0001 (control WT vs severed WT) and $P$ = 0.0816 (severed WT vs severed *p120ctn*). Solid lines indicate one-phase exponential fits. **f** Summary diagram: Anisotropic stress from wing-hinge contraction induces a PD gradient of cell flow. Shear forces−induced via the gradient of cell flow −destabilise transmembrane core proteins (e.g. Fz and Fmi), hindering their accumulation at junctions parallel to the PD-axis. Consequently, this leads to global polarity alignment along the PD-axis. The number of wings examined is indicated. Dot indicates mean, error bars are SEM. Scale bars 10 μm. Source data are provided in the Source Data file.

## Table 1 | Key resources

| Resource | Source | Identifier |
|---|---|---|
| white (w[1118]) | Bloomington Drosophila Stock Center | FBgn0003996 |
| fz::EGFP | 42 | FBti0206968 |
| Sqh::3xmKate | 43,71 | FBal0361068 |
| nub-GAL4 | Bloomington Drosophila Stock Center | FBal0052377 |
| UAS::dpy::RNAi[GD4443] | Vienna Drosophila Stock Center | FBal0209383 |
| fmi::EGFP | 42 | FBtp0137416 |
| E-Cadherin::GFP | 72 | FBal0247908 |
| PH::mCherry | 73 | FBtp0138440 |
| hs-FLP | Bloomington Drosophila Stock Center | FBst0000006 |
| frizzled (fz[P21]) | 74 | FBal0004937 |
| Act > STOP>fz::sfGFP-mKate | 65 | FBal0361851 |
| p120ctn[308] | Bloomington Drosophila Stock Center | FBal0150312 |
| **Software** | | |
| NIS Elements AR version 4.60 | Nikon | N/A |
| Tissue Analyzer | 24 | N/A |
| ImageJ version 2.14.0 | https://fiji.sc | N/A |
| MATLAB_R2016b | MathWorks | N/A |
| GraphPad Prism version 10 | GraphPad software | N/A |
| QuantifyPolarity v2.1 | 44 | N/A |
| **Reagent** | | |
| Halocarbon 700 oil | Halocarbon products | CAS: 9002-83-9 |

EGFP/sfGFP/GFP detection, and a 561 nm laser paired with a 550–595 band-pass filter for mCherry/mKate detection.

The Zeiss LSM880 inverted confocal microscope was equipped with a Plan Apochromat 63×/1.4 NA oil immersion objective. The LSM880 Airyscan detector was used for imaging of red (561 nm laser) and green (a 488 nm laser) channels. All imaging was performed using Immersol 518 F immersion oil (refractive index = 1.518, Carl Zeiss) at a room temperature of 23 °C. Detector gain and pixel dwell times were adjusted for each dataset, keeping them at their lowest values in order to avoid saturation and bleaching effects. Image acquisition utilised the Airyscan detector under consistent laser power settings, and images were processed with the Airyscan processing algorithm in Zen Black software, setting the strength parameter to 6.

Wing specimens were captured in 12-bit z-stacks, consisting of ~20–40 z-slices per stack, with a z-slice interval of 0.5–1 µm. Each image was acquired with a resolution of 512 × 512 pixels and pixel resolution of 0.08–0.14 µm. For all images, unless specified otherwise, the orientation is arranged such that the anterior of the pupal wing is towards the bottom and the proximal side of the pupal wing is towards the left. For long-term time-lapse studies, images were taken at 15-min intervals over ~10 h. Following long-term time-lapse imaging, the pupae were carefully monitored, successfully progressing to at least the pharate stage, with over 95% reaching the eclosion stage.

## Laser ablation experiments

**Circular ablation.** Laser ablation experiments were conducted using a Zeiss LSM880 inverted two-photon microscope equipped with a Chameleon Discovery tunable laser. The laser settings were adjusted to a wavelength of 790 nm, with power levels ranging from 80 to 100%, and utilised a 2× digital zoom in conjunction with a 63× objective lens. Circular ablation was executed with a 5 µm radius at various locations

on the wing blade, specifically below the 5th longitudinal veins and between the 3rd and 4th longitudinal veins from the distal tip of the blade. The ablation process was carried out through the entire z-height of the tissue, ~15 µm, to ensure thorough ablation of the entire monolayer epithelium. For each shot, 30 laser iterations were delivered. In order to capture the recoil of the ablated region, a single z-stack was recorded for a single channel (Fz::EGFP/Fz::sfGFP) with time interval of 60 s.

**Tissue shearing ablation.** Tissue shearing ablation was conducted in the same area as the circular ablation by positioning two rectangles (50 × 5 µm), spaced 70 µm apart, perpendicular to the PD-axis for PD-shear ablation (or parallel for AP-shear). The laser was adjusted to a wavelength of 790 nm, operating at a power range of 80–100%. A series of 30 consecutive laser pulses were applied to the region of interest (ROI) at an approximate z-depth of 15 µm.

**Wing-hinge severing assay.** Pupae were first dissected according to the protocol above. Subsequently, their wings were carefully scratched at the hinge region using a sharp blade, at 20 hAPF.

**Fluorescence recovery photobleaching (FRAP) experiments.** The images obtained were 256 × 256 pixels, with each pixel corresponding to 100 nm with pinhole size of 1.2 AU. Selection for analysis included up to four hub-and-spoke ROIs[46]. Initially, three pre-bleach images were captured at a frequency of 2 frame/s. Following this, the ROIs underwent a bleaching process using a 488 nm Argon laser at 80% power for 8 passes, which lasted a total of 1 s and resulted in a bleaching effect of 70–80%. Directly after bleaching, the imaging sequence was conducted as follows: 5 images were taken every 5 s, followed by 10 images at 10-s intervals, another 10 images at 15-s intervals, and concluding with 8 images at 30-s intervals.

## Quantification and statistical analysis

**Cell polarity, circular weighted histogram and eccentricity analysis.** The 3D raw microscopy image tiles obtained from the imaging were first projected and stitched using the processing pipeline Pre-Mosa to produce a 2D projected image of the apical band of monolayer epithelial tissues[67]. The registered images were segmented using the Tissue Analyzer plugin in Fiji to obtain skeletonised representation of the cell boundaries (also known as segmented images)[24]. All segmented images were checked and corrected manually for segmentation errors. The original and segmented images were then used for quantification of polarity (both magnitude and angle) using QuantifyPolarity[44]. The polarity magnitude and angle of individual cells are calculated using a Principal Component Analysis-based method, which identifies the direction of the greatest variance in weighted intensities from the cell's centroid. When proteins are evenly distributed across cell junctions, the cell is unpolarised. However, if proteins are asymmetrically distributed to opposite junctions (known as bipolarity) the cell becomes polarised with a higher polarity magnitude. The polarity angle is defined as the axis with the greatest asymmetry. Circular weighted histograms were then plotted accordingly using the cell polarity magnitude and angle data obtained from QuantifyPolarity[44]. Data from multiple wings are pooled and divided into 20 bins, with each bin corresponding to a specific range of polarity angles. The histograms are weighted based on the average polarity magnitude, with the radial length indicating the frequency and the direction showing the average polarity angle. All polarity angles are displayed within a range of 0°–360°, with 0°/360° aligned with the horizontal axis of the image. Quantification of cell morphology parameters such as cell eccentricity and cell apical area data were obtained from QuantifyPolarity[44].

**Analysis of tissue stress anisotropy.** First, z-stack images of all time points were projected using the maximum intensity projection algorithm in Fiji. To quantitate anisotropic recoil velocity, an ellipse was fitted to the ablated region, at 120 s after ablation and the major and minor axis lengths were determined using Fiji. Initial recoil velocities of the ablated circle were measured along the semi-major and semi-minor axes respectively, represented by $v_{maj}$ and $v_{min}$ respectively. The orientation of the tissue stress anisotropy was deduced from the orientation of the fitted ellipse and the magnitude of tissue stress anisotropy (which is the anisotropic recoil velocity) was computed as follows,

$$v_a = \frac{v_{maj} - v_{min}}{2} \tag{1}$$

**Quantification of relative stable Fz using fluorescence timer.** The relative amount of stable protein was determined by computing the ratio of total junctional intensities between mKate and sfGFP[68]. To compute relative stable amount of Fz on different junctions: The orientation of each segmented junction was categorised into horizontal and vertical groups based on their orientation relative to the direction of cell flow; horizontal junctions were defined as those with angle spans between −45° and +45° relative to the direction of cell flow and vertical junctions (otherwise). For AP-shear, vertical junctions are defined as those oriented between −45° and +45° relative to the direction of cell flow and horizontal junctions (otherwise). The total junctional intensities for both types of junctions from both imaging channels were extracted accordingly. For each wing image, the relative amount of stable protein was computed using the ratio of mKate to sfGFP total junctional intensities for both horizontal and vertical junctions respectively. It is important to note that variations in imaging conditions, such as laser power, gain, and microscope systems, can affect the ratios of mKate to sfGFP. To ensure the comparability of the relative amounts of stable Fz between experiments, all related experiments were conducted within the same block of sessions using the same microscope and imaging settings.

**Quantification of tissue velocity gradient.** To quantify the relative movements of cells within the tissue, the velocity gradient of the tissue was calculated by adapting a previous approach[60,69] (Supplementary Fig. 4a). Initially, cell identities were tracked throughout the time-lapse using Tissue Analyzer and manually corrected where necessary. This tracking data was then used in a custom MATLAB script[70] to compute the velocity of each cell over time. Subsequently, the imaged region was then divided into a cluster/grid of cells, with grid size selected based on the average cell size. Typically, each grid element contained one to two cell centroids. For coarse-grain tissue velocity gradient analysis (as shown in Supplementary Fig. 4b, d), each grid element could encompass up to four cell centroids. Cells were assigned to specific grid elements based on the proximity of their centroids, with each cell allocated to the nearest grid element that was closest to its centroid. This approach ensured that each grid element represented a localised cluster of cells for accurate tissue velocity gradient analysis.

In the next step, the velocity of the grid elements was computed by averaging the velocity of the cells assigned to each grid element. The velocity of the grid element in the $i$th row and $j$th column, $v(i,j)$, was calculated as

$$v(i,j) = \frac{1}{n_{ij}} \sum_{k}^{n_{ij}} v^{(k)}, \tag{2}$$

where $v^{(k)}$ denotes the velocity of $k$th cell and $n_{ij}$ is the number of cells assigned to this grid element.

For each grid element, velocity gradient tensor D was expressed as:

$$D = \begin{bmatrix} \frac{\partial v_x}{\partial x} & \frac{\partial v_x}{\partial y} \\ \frac{\partial v_y}{\partial x} & \frac{\partial v_y}{\partial y} \end{bmatrix}, \tag{3}$$

where each component represents the velocity gradient in a specific direction. Using the finite difference method, each of the velocity gradient components was approximated as follows:

$$\frac{\partial v_x}{\partial x}(i,j) \approx \frac{\Delta v_x}{\Delta x} = \frac{v_x(i,j+1) - v_x(i,j)}{x(i,j+1) - x(i,j)} \tag{4}$$

$$\frac{\partial v_y}{\partial x}(i,j) \approx \frac{\Delta v_y}{\Delta x} = \frac{v_y(i,j+1) - v_y(i,j)}{x(i,j+1) - x(i,j)}, \tag{5}$$

$$\frac{\partial v_x}{\partial y}(i,j) \approx \frac{\Delta v_x}{\Delta y} = \frac{v_x(i+1,j) - v_x(i,j)}{y(i+1,j) - y(i,j)} \tag{6}$$

$$\frac{\partial v_y}{\partial y}(i,j) \approx \frac{\Delta v_y}{\Delta y} = \frac{v_y(i+1,j) - v_y(i,j)}{y(i+1,j) - y(i,j)} \tag{7}$$

where $i$ and $j$ denote the row and column indices of the grids respectively.

To account for conditions where the cell flow direction is not aligned with the $x$- or $y$-axis, the coordinate system was rotated to match the cell flow direction. The direction of cell flow, denoted as $[\bar{v}_x, \bar{v}_y]^T$, was calculated based on the nematic average velocity vector across all cells over the observation period and is defined as:

$$\begin{bmatrix} \bar{v}_x \\ \bar{v}_y \end{bmatrix} = \frac{1}{n} \sum_{k=1}^{n} |v_k|^2 \cdot \begin{bmatrix} \cos(2\varphi_k) \\ \sin(2\varphi_k) \end{bmatrix} \tag{8}$$

where $|v_k|$ and $\varphi_k$ represent the magnitude and angle of the velocity for the $k$th cell. The angle $\theta$, indicating the direction of cell flow relative to the $x$- or $y$-axis, was computed as follows:

$$\theta = \begin{cases} \tan^{-1}\left(\frac{\bar{v}_y}{\bar{v}_x}\right) & \text{for all conditions except AP} - \text{shear,} \\ \tan^{-1}\left(\frac{\bar{v}_x}{\bar{v}_y}\right) & \text{for AP} - \text{shear.} \end{cases} \tag{9}$$

The velocity gradient tensor in the rotated coordinate system, $\mathbf{D}'$, was then obtained using the tensor transformation law:

$$D' = R(\theta) \cdot D \cdot R^{-1}(\theta) \tag{10}$$

where $R(\theta)$ is the 2D rotation matrix defined as:

$$R(\theta) = \begin{bmatrix} \cos\theta & -\sin\theta \\ \sin\theta & \cos\theta \end{bmatrix} \tag{11}$$

This process was applied to each grid to calculate its respective velocity gradient. Subsequently, the velocity gradients across all grids within the imaged region were aggregated to yield the overall tissue-wide velocity gradient, reflecting a tissue-wide measurement of the velocity gradient:

$$\frac{\partial v_x}{\partial y} = \frac{1}{(n_i - 1) \cdot (n_j - 1)} \sum_{j=1}^{n_j-1} \sum_{i=1}^{n_i-1} \left| \frac{\partial v_x}{\partial y}(i,j) \right| \tag{12}$$

where $n_i$ and $n_j$ represent the total number of rows and columns respectively. Since the values of $\frac{\partial v_x}{\partial x}$ and $\frac{\partial v_y}{\partial y}$ showed no significant

differences across conditions and no correlation with Fz stability, these components were considered negligible (Supplementary Fig. 4h, h').

The instantaneous velocity gradient measure was calculated at 15-min intervals to provide detailed information about how velocity changed at each specific time point, capturing rapid fluctuations in the dynamics within the tissue. Meanwhile, the average velocity gradient was determined by averaging these instantaneous velocity gradients over a 2-h period, providing a broader perspective on the changes in the velocity field throughout the experiment.

**Quantification of Fz::EGFP junctional enrichment and length during T1 transitions.** To examine Fz protein accumulation at newly formed junctions following T1 transitions, long-term time-lapse imaging of Fz::EGFP was performed at 15-min intervals from 24 to 36 hAPF. All T1 transition events, characterised by the contraction and extension of cell-cell junctions in the apical plane (as depicted in Fig. 3a), occurring between 24 and 36 hAPF were manually identified. A total of at least 100 T1 transition events were recorded for each genotype/condition. The mean junctional intensities and lengths of these newly extended junctions (horizontal or vertical junctions) were monitored and extracted for ~180 min post-formation using a line (pixel width 3) function tool in ImageJ. Analysis focused on new extended junctions that remained stable over time without reverting to their original configuration. The mean intensities/length of these new junctions were then categorised according to orientation of the junction (horizontal or vertical). The percentage of change in Fz density was calculated as below:

$$\text{\%change in Fz density} = \frac{I_t - I_0}{I_0} \times 100 \qquad (13)$$

where $I_0$ and $I_t$ represents Fz density at time 0 (immediately after new junction formation following 4-way vertex dissolution) and time $t$ respectively.

These values were further averaged across each wing for comparative analysis. Similarly, the percentage of change in junction length was computed as follows:

$$\text{\% change in junction length} = \frac{l_t - l_0}{l_0} \times 100 \qquad (14)$$

where $l_0$ and $l_t$ represents junction length at time 0 (immediately after new junction formation following 4-way vertex dissolution) and time $t$ respectively. These values were also further averaged across each wing for comparative analysis.

**Quantification of Fz::EGFP stability using FRAP.** ROI for bleached regions was manually drawn in each image for each time point using the Bezier function in ImageJ. The mean intensity within each ROI was measured using ImageJ. After subtracting the laser off background intensity, acquisition bleaching and the data were normalised to the average pre-bleach values. These normalised values were graphically represented on an XY plot using GraphPad Prism software, where one-phase exponential curves were applied to examine for the goodness of fit. Curves were excluded if the ROI recovery curve did not pass Prism's 'replicates test for lack of fit'. The values from multiple bleached regions were averaged per wing. Subsequently, data from multiple wings were aggregated, and one-phase exponential association curves were fitted to the data.

**Quantification of Fz density and stability during relative cell movement.** To examine changes in Fz density during relative cell movement, time-lapse imaging of Fz::EGFP was performed at 15-min intervals from 24 to 28 hAPF. All relative cell movement events, characterised by the loss and gain of junctions of similar orientation (as depicted in Fig. 4d), occurring between 24 and 28 hAPF were manually identified. A total of at least 70–100 relative cell movement events were tracked for each genotype/condition. The mean intensities of shortening and extending junctions were tracked and extracted for ~120 min post-formation using the line tool (pixel width 3) in ImageJ. These mean intensities were normalised to those of other junctions within the same cell. A value below 1 indicates that the mean intensity of the junction of interest is lower compared to that of the cell's other junctions. Conversely, a value above 1 indicates that the mean intensity of the junction of interest is higher than that of the cell's other junctions. On the other hand, the percentage change in Fz density for extending junctions was computed at 60 and 120 min post extension as above (under 'Quantification of Fz::EGFP junctional enrichment and length during T1 transitions') and averaged across each wing for further comparative analysis.

To compute changes in relative stable amount of Fz on sliding junction during relative cell movement event (Fig. 7d), the mean intensities of both Fz::sfGFP and Fz::mKate on sliding junctions before and after relative cell movement were determined using the line tool (pixel width 3) in ImageJ. The relative amount of stable protein on the sliding junctions pre- and post-movement, denoted as $S_{pre}$ and $S_{post}$, were then determined by computing the ratio of the mean junctional intensities of mKate to sfGFP, as described below:

$$S_{pre} = \frac{\sum I_{Fz::mKate}}{\sum I_{Fz::sfGFP}} \qquad (15)$$

$$S_{post} = \frac{\sum I_{Fz::mKate}}{\sum I_{Fz::sfGFP}} \qquad (16)$$

Then, the percentage of change in Fz stability on sliding junctions pre- and post-movement was computed as follows:

$$\text{\% change in Fz stability} = \frac{S_{post} - S_{pre}}{S_{pre}} \times 100 \qquad (17)$$

**Statistical analysis.** Statistical analysis was performed using GraphPad Prism v10. All quantification measures such as polarity magnitude, polarity angle variance, anisotropic recoil velocity, cell eccentricity, relative stable amount of Fz and tissue velocity gradient for multiple genotypes were averaged per wing and tested for normality with the Shapiro-Wilk test. When the distribution was normal, ANOVA or Student's $t$-test was performed to estimate significance of the quantities. In case the values were not normally distributed, non-parametric Mann–Whitney $U$-test was performed to estimate the significance of differences between the quantities. For pupal wings, each experiment was performed on multiple wings from different pupae, which represent biological replicates ($n$ = number of wings). The resulting data from multiple wings were compared using an unpaired one-way ANOVA test or paired/unpaired Student's $t$-test using GraphPad Prism. All statistical tests are two-sided. For one-way ANOVA test, if it was statistically significant, a following post hoc Tukey–Kramer's multiple comparison was used to compare all genotypes of the same developmental time point within an experiment. The corresponding $P$ values and the method used to estimate them are mentioned in the figure legends. In all graphs, error bars represent Standard Error of Mean (SEM). In some experiments, coefficient of determination test ($R^2$) was computed to determine how strong the linear relationship was between two variables using GraphPad Prism. For FRAP analysis, $P$ values were obtained by comparing the fits of two datasets based on the plateau parameter, using an extra sum-of-squares $F$-test.

## Reporting summary

Further information on research design is available in the Nature Portfolio Reporting Summary linked to this article.

## Data availability

Source data supporting the findings of this study derived from quantitative analysis of microscope images are provided with this paper in the Source Data file. The primary image data that support the findings of this study are too large to deposit in a public repository but are available from the corresponding author upon request. Source data are provided with this paper.

## Code availability

The source code for tissue velocity gradient analysis is available in GitHub (https://doi.org/10.5281/zenodo.14592763).

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

## Acknowledgements

We extend our gratitude to Kyra Campbell for granting us access to her microscope for laser ablation experiments. Our special thanks go to Strutt lab members for their invaluable insights. We are grateful to Weijie Tan for guidance on analysing tissue flow and Helen Strutt for comments on the manuscript. Special thanks to the University of Sheffield's Wolfson Light Microscopy Facility and the Fly Lab for excellent technical support. This work was funded by a Wellcome Trust Senior Fellowship (210630/Z/18/Z) and an EPSRC grant (EP/W024144/1) to D.S.

## Author contributions

S.T. designed the research, performed experiments, wrote the code, analysed the data and wrote the manuscript. D.S. designed the research and edited the manuscript.

## Competing interests

The authors declare no competing interests.
