## [Peer Review file · Nature Communications]

Tissue shear as a cue for aligning planar polarity in the developing *Drosophila* wing

Corresponding Author: Professor David Strutt

Version 0:

Reviewer comments:

Reviewer #1

(Remarks to the Author)

In this manuscript the authors investigated the relationship between mechanical stress and/or velocity gradient of cells (shear) and Fz instability on cellular membrane. With disturbance of stress by Dumpy-RNAi, severing the wing from the hinge, and laser ablation, the authors monitored Fz dynamics before and after T1 transition. The authors claims that shear gradient causes decrease in Fz on the cellular membrane, that coordinates long-ranged polarization of PCP.

I agree that the coordination of PCP by mechanical stress is a important issue for understanding the morphogenesis, and appreciate their thorough investigation, but I feel that the results are rather descriptive and interpretation of the experimental results for normal developmental process is unclear, mainly by the lack of molecular mechanism of the Fz accumulation process.

1. The authors mention 'rows' and 'columns' of cells in the main text, but it is less clear how the direction of the cell array is determined. The defeminization procedure of the coordinate is very important because shear deformation, in particular the measurement 'velocity_x gradient' can depend on the choice of the direction. I found some descriptions in Materials and Methods, but I am not yet convinced because of the lack of enough information such as the magnitude of the other components of the velocity gradient tensor.

2. The authors mainly investigated the sliding process of cells, that is different from the typical T1 transition. As far as I understand, although the sliding process may occur in a part of the developing *Drosophila* wing, the typical T1 process is a major mode of cell position change in the tissue. How is a sliding process important for tissue morphogenesis? The stress distribution is different between the two modes? Does the molecular mechanism of the Fz accumulation/destabilization differ between these two modes?

3. In the cell sliding process, do cells rotate or not? I am asking the relative position of contacting cellular membranes itself slide or not, which could be important for molecular machinery of Fz accumulation.

4. In Fig. 5 and Fig. 6, the velocity gradient is 'acutely reduced' by laser ablations in the wing. The induced gradients seem sufficiently high (Fig. 5b), however, it is a response to the ablation and thus may be transient and occur in short time. In the response, do cells show sliding motion accompanying relative positional change of cells (T1 transition)? Is the response of Fz to the velocity gradient caused by such a transient response considered the same as in normal development, which proceeds over a longer period of time?

5. What is the molecular mechanism of Fz destabilization after T1 transition? Is it associated with increase of PTEN (Bardet et al. Dev Cell 2013, for 'typical T1' transition) or detachment of myosin from cellular membrane during T1 (Ikawa et al. Curr. Biol. 2023) ?

Reviewer #2

(Remarks to the Author)

This manuscript describes effects of tissue stress on planar cell polarity (PCP) alignment in the *Drosophila* wing. It builds on

(and partially reproduces) early work of Aigouy et al (2010), who reported that shear reorients PCP during pupal wing development. What this manuscript adds is more insight into the mechanism. By directly examining Fz as a molecular readout for core PCP complexes and using a fluorescent timer construct to assess Fz stability, and correlating effects on Fz with shear stress and the behavior of cells and cell junctions, the authors obtain results that lead them to propose that the main factor reorienting PCP is the disruption of intercellular complexes by shear between neighboring cells. The results provide new insights into how PCP is aligned during development that should be of broad interest. However, there are some issues that need to be addressed.

Major Concerns:

1) There is a major problem with some of the analysis in dpy RNAi, as the authors compare distal regions of dpy to proximal regions of control, when there are substantial differences between distal and proximal regions. This is not acceptable. This occurs in results presented in Fig. 2 and Fig S1. In Figures 3 and 4 it's not clear whether the comparison between dpy results and control results are valid, as it's not clear what regions of the wing were used. This needs to be clarified, and it's essential that the same regions be compared.

2) In Fig 6b, the authors report rotational cell trajectories in dpy RNAi, which they call surprising, but they never show us control wings at this age. The wild-type control at this age also needs to be analyzed here.

Minor issues:

1) It would be helpful if the authors could relate their analysis of cell flows and Fz orientation in pupal wings to PCP patterns in adult wings, eg describe or refer to analysis of the PCP pattern in dpy RNAi.

2) In Fig. 4j, it would be more meaningful if the authors compared Fz stability of horizontal versus proximal junctions, as they did in Fig. 3.

3) In Figure 5, the authors describe an experiment that seems not to have worked, so it's not clear that it makes sense to include it. I'm referring to the AP-shear experiment, which didn't show the anticipated differences between horizontal and vertical junctions, which the authors then explain by saying maybe the velocity gradient was too small. If this is the case, then perhaps they should be devising an alternative experiment?

4) The quantitation for some of the data is unclear. eg How many junctions are represented by the data shown in Fig 3C?

5) While the authors have nice explanatory diagrams for some of the data, some of the analysis is complex and could be explained better.

- A summary figure at the end depicting their main conclusion would be helpful.

- It would also be helpful if the orientation of horizontal versus vertical junctions relative to the pupal wing could be included in an explanatory diagram, as this terminology is not intuitive.

- In Figures 4 and S5, it would be helpful to have a schematic for the Velocity profile (row and column) in reference to the pupal wing.

- The dots in Fig 4k are so small that it's hard to follow the color scheme.

- In Figure 1 (& some subsequent figures), the boxes on the wing are a different shape (rectangular) from the image boxes (square), even though they are supposed to be indicating where the image boxes are from. This should be fixed as it's confusing. It would also be nice to have orientation of the image boxes clearly indicated.

- In Fig S4, boxes should be placed on the wing schematic to show us what regions the data being shown comes from.

Reviewer #3

(Remarks to the Author)

I co-reviewed this manuscript with one of the reviewers who provided the listed reports as part of the Nature Communications initiative to facilitate training in peer review and appropriate recognition for co-reviewers.

Reviewer #4

(Remarks to the Author)

Tissue stress orients PCP in both invertebrate and vertebrate epithelia, which is at least partly explained by the slow accumulation of PCP core proteins along nascent junctions (Aigouy et al. Cell, 2010; Wu et al. Curr. Biol., 2016). However, the mechanistic details of this process remain unclear.

Tan and Strutt addressed the question by examining the formation of Fz polarity in *Drosophila* pupal wing. They demonstrated that tissue stress anisotropy inversely correlates with Fz stability. Moreover, they observed a correlation between tissue stress anisotropy and differential velocity along cell rows. The acute induction of differential cell flow is sufficient to reorient the Fz polarity in dpy RNAi wings. Finally, they showed that Fz and Fmi exhibit lower stability along sliding junctions, which result from differential cell flow. Overall, this study represents an important addition to the field and is worthy of publication after addressing the points below.

Major points

1. Fig. 2. Data acquired in different regions of the wing are compared between control and dpy RNAi wings.

2. I do not understand why Fz accumulates on newly formed horizontal junctions but not on nearby existing horizontal junctions in Fig. S3, given that both are exposed to similar magnitudes of tissue stress and cell flow gradients at the time of observation. Does the majority of nearby existing horizontal junctions undergo sliding?
3. Fig. 4e–g. The difference in the velocity gradient between the distal part of the control wing and the *dpy* RNAi wing is relatively small; yet, the Fz polarity at 32 h APF is severely disrupted in the *dpy* RNAi wing. Please discuss this point.
4. Tracking the Fz polarity and stability in the same cells would provide further support for the authors' claims. Such analysis could clarify how significantly the mechanism proposed here contributes to the formation of Fz polarity.
5. Fig. 4k. Could you please enlarge the plot for better visibility? Are the velocity gradient values plotted in Fig. 4k consistent with those shown in Fig. 4e–g?
6. Fig. 7 and Fig. S7. Does manipulating cell adhesion strength affect Fz stability and polarity?
7. p. 12. Ref. 51 and 52 consider active cell rearrangement as a mechanism for inducing tissue shear; however, the cell rearrangement during the developmental stage studied in this manuscript is passively induced as a means to relax tissue stretch. Additional factors, such as the stiffness of veins and wing margin as boundary conditions, as well as spatial variations in cell mechanical/geometrical properties, can be involved. The link between tissue stress and cell flow, therefore, needs to be discussed more carefully. I listed some potentially relevant literature below: Ray et al., *Cell*, 2015 (Ref. 50), which conducts numerical simulations with different tensions at the vein-intervein boundary; Ishimoto and Sugimura. *J Theor. Biol.*, 2017, regarding the inference of wing margin stiffness; Claussen et al., arXiv, 2024, which describes numerical simulations of patterned flow. I believe there are many more relevant studies.
8. Methods. The authors first determined the row of cells based on average cell height and then computed the velocity for each cell row. Please describe how cells undergoing T1 or T2 transitions are treated. I wonder why they did not quantify the velocity field without separating cells into rows and calculate the velocity gradient from the velocity field.

Minor points

1. Methods. Please provide the definition of the polarity magnitude quantified using QuantifyPolarity (Ref. 44).
2. Please set the minimum value of the y-axis to zero to accurately represent the data, except for the stable fraction of Fz, where cells should maintain a certain value for junctional localization.
3. Fig. 3a. Yellow and blue in the second schematic in the second row are swapped.
4. Fig. 3f, g and other figures. Please specify the region of the wing from which the data were collected.
5. Fig. 4 and 5. Why is the AP-axis inverted?
6. Fig. 4k. R^2 is shown as 0.7923 in the figure, whereas it is reported as 0.7668 in the main text. Please clarify.
7. Fig. 5 and Fig. 6. Line ablation halts tissue flow rather than promoting it in the *Drosophila* embryo, as reported by Collinet et al. (*Nat. Cell Biol.*, 2015). Please discuss this point.
8. Fig. S1e. Labeling is missing.
9. Ref. 56 is not complete.

Version 1:

Reviewer comments:

Reviewer #1

(Remarks to the Author)

The revised manuscript shows significant improvement, with the addition of some excellent figures and discussions. The authors' claims are now clearer, and their responses to my comments are satisfactory. I agree that exploring the molecular mechanism is beyond the scope of this work and should be addressed in future studies. I now recommend the manuscript for publication in *Nature Communications*.

I would like the author to clarify the following points before the publication.

1. I am still confused about the relationship between sliding and shear. In your laser ablation experiment, do cell membranes slide against each other? The observed shear could be attributed to cell shape changes with minimal cell membrane sliding.

Minor points:

1. Some figures (plots and histograms) are small, making it difficult to see the characters.
2. L375: Fig. 4a,b, 4f should be changed to Fig. 4a,b,f.
3. L399: A period is missing.
4. L400: The figure number is missing.

(Remarks on code availability)

Reviewer #2

(Remarks to the Author)

The authors revisions have addressed the concerns raised - it's an important addition to the field.

(Remarks on code availability)

Reviewer #3

(Remarks to the Author)

(Remarks on code availability)

Reviewer #4

(Remarks to the Author)

The authors have greatly improved the manuscript by performing a series of new experiments and data analyses. I agree with the publication of this study.

Below are some minor comments that I hope will be helpful:

- Fig. 1c. Replace v_{maj} with v_{max} .
- Fig. 8. I would suggest replacing "dynamics" with "turnover," as the former covers wider aspects, such as lateral clustering.
- Fig. S3a. The remodeling junction in the second panel of the top row is not a 4-way vertex. I would suggest using the same labeling as in Fig. 3.
- Fig. S4g and g'. These panels demonstrate the correlation between the tissue velocity gradient and Fz stability. They are worth including in the main text if space allows.
- Line 346–350. The ref. 51 and 52 in the original manuscript (Sato et al., 2015; Popovic et al., 2017) are missing from the bibliography of the revised manuscript. It seems that there are inconsistencies between the main text and the bibliography.
- Line 400. The figure number is missing.

Below are some discussions that could be informative for the authors' future studies:

- The MT growth polarity varies along the PD axis (Harumoto et al., 2010), similar to the spatial variation in the tissue flow gradient that the authors demonstrated in this study. I wonder if these two are regulated by a common mechanism. If so, are they chemically prepatterned or a consequence of mechanotransduction?
- As the authors assumed in their response to reviewer 1, tissue stress anisotropy was unaffected by RNAi of *aip1* or *cofilin* (Fig. S4 in the reference).

Kaoru Sugimura

(Remarks on code availability)

Response to reviewers

We thank the reviewers for their detailed and thoughtful comments. In response we have carried out more analysis, performed further experiments and made numerous revisions to the manuscript (which are highlighted in the resubmitted version).

Details are provided below.

Reviewer #1

In this manuscript the authors investigated the relationship between mechanical stress and/or velocity gradient of cells (shear) and Fz instability on cellular membrane. With disturbance of stress by Dumpy-RNAi, severing the wing from the hinge, and laser ablation, the authors monitored Fz dynamics before and after T1 transition. The authors claims that shear gradient causes decrease in Fz on the cellular membrane, that coordinates long-ranged polarization of PCP. I agree that the coordination of PCP by mechanical stress is a important issue for understanding the morphogenesis, and appreciate their throughfall investigation, but I feel that the results are rather descriptive and interpretation of the experimental results for normal developmental process is unclear, mainly by the lack of molecular mechanism of the Fz accumulation process.

In response to the concern that our results are 'descriptive' and lack clear relevance to normal developmental processes, we would like to argue that our experiments extend beyond just description. We actively manipulated mechanical stress through various methods, such as dpy-RNAi, hinge severing, and laser ablation. These interventions impact Fz stability and polarity, as demonstrated by changes in Fz dynamics on the cellular membrane. By directly altering tissue stress anisotropy and observing the resulting changes in core proteins, we establish a clear connection between mechanical forces and developmental polarity cues.

While the molecular mechanisms underlying Fz accumulation/destabilisation are not fully elucidated, we do discuss possible mechanisms, such as the shearing of transmembrane complexes, in the context of our findings. Understanding how shear affects molecular complexes is indeed an important question, but it is beyond the scope of this work, which focuses on developmental patterning and the mechanobiological regulation of PCP.

1. The authors mention 'rows' and 'columns' of cells in the main text, but it is less clear how the direction of the cell array is determined. The defemination procedure of the coordinate is very important because shear deformation, in particular the measurement 'velocity_x gradient' can depend on the choice of the direction. I found some descriptions in Materials and Methods, but I am not yet convinced because of the lack of enough information such as the magnitude of the other components of the velocity gradient tensor.

To clarify how the tissue velocity gradient is computed, we have included additional information and visual aids (e.g. flowcharts and diagrams) in the revised manuscript (new Supplementary Figure 4a-b). We further validated our approach using the triangulation method by Etournay et al. (2015, <https://doi.org/10.7554/eLife.07090>) for measuring tissue shear rate, which corresponds to the tissue velocity gradient in our study. Notably, the tissue shear rate obtained using the triangulation method gave similar results to our tissue velocity

gradient measurements across all genotypes at 24 hAPF (compare Fig. 4f and Supplementary Fig. 4c).

We acknowledge the reviewer's concern regarding the lack of information for the magnitude of other velocity gradient tensor components. Apart from velocity_x gradient and velocity_y gradient measurements as shown in Figure 4 and Supplementary Figure 5, we also measured the magnitudes of two other velocity gradient tensor components (namely $\frac{\partial v_x}{\partial x}$ and $\frac{\partial v_y}{\partial y}$). Since the values of $\frac{\partial v_x}{\partial x}$ and $\frac{\partial v_y}{\partial y}$ showed no significant differences across conditions and showed no correlation with Fz stability, these components were considered negligible (new Supplementary Fig. 4h-h'). Similarly, as shown in Supplementary Figure 5, the velocity_y gradient was significantly lower than the velocity_x gradient and showed no correlation with Fz stability. Our analysis of wild-type wings indicates that the highest velocity gradient is along the PD-axis, corresponding to the velocity_x gradient. This demonstrates that the velocity_x gradient is the most prominent factor with the strongest correlation to Fz instability.

We have included more details of the analysis in the Materials & Methods of the revised manuscript as below:

Quantification of tissue velocity gradient

To quantify the relative movements of cells within the tissue, the velocity gradient of the tissue was calculated by adapting a previous approach [Smutny et al 2017 (doi: 10.1038/ncb3492) and Blanchard et al 2009 (doi: 10.1038/nmeth.1327)] (Supplementary Fig. 4a). Initially, cell identities were tracked throughout the timelapse using Tissue Analyzer and manually corrected where necessary. This tracking data was then used in a custom MATLAB script to compute the velocity of each cell over time. Subsequently, the imaged region was then divided into a cluster/grid of cells, with grid size selected based on the average cell size. Typically, each grid element contained one to two cell centroids. For coarse-grain tissue velocity gradient analysis (as shown in Supplementary Fig. 4b, 4d), each grid element could encompass up to four cell centroids. Cells were assigned to specific grid elements based on the proximity of their centroids, with each cell allocated to the nearest grid element that was closest to its centroid. This approach ensured that each grid element represented a localised cluster of cells for accurate tissue velocity gradient analysis.

In the next step, the velocity of the grid elements was computed by averaging the velocity of the cells assigned to each grid element. The velocity of the grid element in the *i*-th row and *j*-th column, $v(i, j)$, was calculated as

$$v(i, j) = \frac{1}{n_{ij}} \sum_k^{n_{ij}} v^{(k)},$$

where $v^{(k)}$ denotes the velocity of *k*-th cell and n_{ij} is the number of cells assigned to this grid element.

For each grid element, velocity gradient tensor **D** was expressed as:

$$\mathbf{D} = \begin{bmatrix} \frac{\partial v_x}{\partial x} & \frac{\partial v_x}{\partial y} \\ \frac{\partial v_y}{\partial x} & \frac{\partial v_y}{\partial y} \end{bmatrix},$$

where each component represents the velocity gradient in the specific direction. Using the finite difference method, each of the velocity gradient components was approximated as follows:

$$\begin{aligned}\frac{\partial v_x}{\partial x}(i, j) &\approx \frac{\Delta v_x}{\Delta x} = \frac{v_x(i, j+1) - v_x(i, j)}{x(i, j+1) - x(i, j)}, \\ \frac{\partial v_y}{\partial x}(i, j) &\approx \frac{\Delta v_y}{\Delta x} = \frac{v_y(i, j+1) - v_y(i, j)}{x(i, j+1) - x(i, j)}, \\ \frac{\partial v_x}{\partial y}(i, j) &\approx \frac{\Delta v_x}{\Delta y} = \frac{v_x(i+1, j) - v_x(i, j)}{y(i+1, j) - y(i, j)}, \\ \frac{\partial v_y}{\partial y}(i, j) &\approx \frac{\Delta v_y}{\Delta y} = \frac{v_y(i+1, j) - v_y(i, j)}{y(i+1, j) - y(i, j)},\end{aligned}$$

where i and j denote the row and column indices of the grids respectively.

To account for conditions where the cell flow direction is not aligned with the x- or y-axis, the coordinate system was rotated to match the cell flow direction. The direction of cell flow, denoted as $[\bar{v}_x, \bar{v}_y]^T$, was calculated based on the nematic average velocity vector across all cells over the observation period and is defined as:

$$\begin{bmatrix} \bar{v}_x \\ \bar{v}_y \end{bmatrix} = \frac{1}{n} \sum_{k=1}^n |v_k|^2 \cdot \begin{bmatrix} \cos(2\varphi_k) \\ \sin(2\varphi_k) \end{bmatrix},$$

where $|v_k|$ and φ_k represent the magnitude and angle of the velocity for the k -th cell. The angle θ , indicating the direction of cell flow relative to the x- or y-axis, was computed as follows:

$$\theta = \begin{cases} \tan^{-1}\left(\frac{\bar{v}_y}{\bar{v}_x}\right) & \text{for all conditions except AP - shear,} \\ \tan^{-1}\left(\frac{\bar{v}_x}{\bar{v}_y}\right) & \text{for AP - shear.} \end{cases}$$

The velocity gradient tensor in the rotated coordinate system, \mathbf{D}' , was then obtained using the tensor transformation law:

$$\mathbf{D}' = \mathbf{R}(\theta) \cdot \mathbf{D} \cdot \mathbf{R}^{-1}(\theta),$$

where $\mathbf{R}(\theta)$ is the 2D rotation matrix defined as:

$$\mathbf{R}(\theta) = \begin{bmatrix} \cos \theta & -\sin \theta \\ \sin \theta & \cos \theta \end{bmatrix}.$$

This process was applied to each grid to calculate its respective velocity gradient. Subsequently, the velocity gradients across all grids within the imaged region were aggregated to yield the overall tissue-wide velocity gradient, reflecting a tissue-wide measurement of the velocity gradient:

$$\frac{\partial v_x}{\partial y} = \frac{1}{(n_i - 1) \cdot (n_j - 1)} \sum_{j=1}^{n_j-1} \sum_{i=1}^{n_i-1} \left| \frac{\partial v_x}{\partial y}(i, j) \right|,$$

where n_i and n_j represent the total number of rows and columns respectively. Since the values of $\frac{\partial v_x}{\partial x}$ and $\frac{\partial v_y}{\partial y}$ showed no significant differences across conditions and no correlation with Fz stability, these components were considered negligible (Supplementary Fig. S4h-h').

The instantaneous velocity gradient measure was calculated at 15-minute intervals to provide detailed information about how velocity changed at each specific time point, capturing rapid fluctuations in the dynamics within the tissue. Meanwhile, the average velocity gradient was determined by averaging these instantaneous velocity gradients over a 2-hour period, providing a broader perspective on the changes of the velocity field throughout the experiment.

New Supplementary Figure 4a-c

(a,b) Flowchart illustrating the steps for quantifying tissue velocity gradient at both local (coarse-grain) and global (average) levels. Each timelapse is captured, and all cells are segmented and manually tracked over time. The imaged region is then divided into smaller grids or clusters, with each grid containing approximately 1-4 cells (as shown by the square boxes in (b)). Cells within each grid are tracked over time to determine the average velocity of the specific grid. The velocity gradient between adjacent grids is calculated using the finite difference method, and this process is repeated across all grids to obtain the coarse-grain velocity gradient. The global velocity gradient is calculated by averaging all the coarse-grain velocity gradients.

(c) Validation of the tissue velocity gradient method against the published triangulation method for velocity gradient computation across all genotypes at 24 hAPF. $n = 3-4$ wings per genotype. One-way ANOVA test, comparing WT proximal and *dpy-RNAi* wings to WT distal. *** $P = 0.0004$, * $P = 0.0486$.

New Supplementary Figure 4h-h'

(h,h') Quantification of velocity gradient tensor components (h) $\frac{\partial v_x}{\partial x}$ and (h') $\frac{\partial v_y}{\partial y}$ for proximal and distal regions of WT wings and *dpy-RNAi* wings at 24 hAPF. $n = 3$ timelapses per genotype. One-way ANOVA unpaired test, comparing WT proximal and *dpy-RNAi* wings to WT distal wings. ns, not significantly different. Dot indicates mean per wing, error bars are SEM.

2. The authors mainly investigated the sliding process of cells, that is different from the typical T1 transition. As far as I understand, although the sliding process may occur in a part of the developing *Drosophila* wing, the typical T1 process is a major mode of cell position change in the tissue. How is a sliding process important for tissue morphogenesis? The stress distribution is different between the two modes? Does the molecular mechanism of the Fz accumulation/destabilization differ between these two modes?

We thank the reviewer for their insightful comments and the opportunity to clarify our findings regarding the sliding process of cells in comparison to the typical T1 transition.

In general, sliding processes are known to play a significant role in tissue morphogenesis. Shear forces arising from cell sliding play a critical role in determining the position of the neural anlage during zebrafish embryogenesis (Smutny et al., 2017 <https://doi.org:10.1038/ncb3492>). Besides that, cell sliding has been shown to be critical for blood vessel branch elongating during angiogenic morphogenesis (Tonami et al., 2023 <https://doi.org:10.1016/j.isci.2023.107051>). Our findings support the idea that the sliding of cells constitutes a key global tissue-level cue for pattern formation in development. We are unsure whether the stress distribution differs between the two modes, but we speculate that both sliding and T1 transitions may serve as stress dissipative mechanisms that help relieve stress within the system, thereby maintaining tissue integrity.

In direct response to the reviewer's comment we have now investigated whether the effect of relative cell movement on Fz stability might be mediated through altered rates of T1 transitions. We tracked the frequency of T1 transitions in our laser ablation-induced PD-shear region (Fig. 5) and compared them to the non-ablated region. Acute tissue ablation led to fewer T1 transition events compared to the unablated control region (Supplementary Fig. 6g), despite the observed lower Fz stability (Fig. 5d). We also found that the distal wild-type wing region has fewer T1 transitions compared to the proximal region (Supplementary Fig. 6g'), but in this case correlating with higher Fz stability (Fig. 1h). These findings suggest that Fz stability is not directly influenced by the rate of T1 transitions.

We have included this in the revised manuscript as below:

See Results section: Acute induction of a tissue velocity gradient results in reduced Fz stability and altered polarity orientation

We next asked whether the effect of relative cell movement on Fz stability might be mediated through altered rates of T1 transitions. We tracked the frequency of T1 transitions in the PD-shear region and compared them to the non-ablated region. Acute tissue ablation led to fewer T1 transition events compared to the unablated control region (Supplementary Fig. 6g), despite the observed lower Fz stability (Fig. 5d). We also found that the distal wild-type wing region had fewer T1 transitions compared to the proximal region (Supplementary Fig. 6g'), but in this case correlating with higher Fz stability (Fig. 1h). These findings suggest that Fz stability is not directly influenced by the rate of T1 transitions.

New Supplementary Figure 6g-g'

(g) Quantification of frequency of T1 transition for both control and PD-shear regions. $n = 3$ wings. Unpaired t-test, $*P = 0.0179$.

(g') Quantification of frequency of T1 transition for proximal and distal regions of WT wings at 28 hAPF. $n = 3$ wings. Unpaired t-test, $*P = 0.0402$.

3. In the cell sliding process, do cells rotate or not? I am asking the relative position of contacting cellular membranes itself slide or not, which could be important for molecular machinery of Fz accumulation.

Currently, we have no evidence that cells rotate during the sliding process. An in vitro study has shown that two cultured cells can undergo both linear movement and rotational movement relative to each other (Tonami et al., 2023), and demonstrated that these movements could coexist in a two-cell state. However, in a sheet of confluent cells, rotation is not evident, with cells simply moving linearly in opposite directions. The rotational movement observed in the two-cell state is likely due to the lack of adhesion to other neighbouring cells. Evidence further suggests that the loss of VE-Cadherin enhances rotational movement, indicating that loosely adhered cells are more prone to this behaviour. In contrast, we speculate that cell rotation may be more energetically costly in tightly packed tissues, as E-Cadherin would need to continually attach, detach, and reattach.

While it is possible that cell rotation could reduce slip/shear between contacting horizontal membranes—depending on its angular and linear velocity—this might lead to slip or shear on other contacting membranes, leading to more general destabilisation of Fz. However, it is not obvious how such a mechanism would contribute to specifically promoting PD polarisation.

4. In Fig. 5 and Fig. 6, the velocity gradient is 'acutely reduced' by laser ablations in the wing. The induced gradients seem sufficiently high (Fig. 5b), however, it is a response to the ablation and thus may be transient and occur in short time. In the response, do cells show sliding motion accompanying relative positional change of cells (T1 transition)? Is the response of Fz to the velocity gradient caused by such a transient response considered the same as in normal development, which proceeds over a longer period of time?

We thank the reviewer for their insightful questions regarding the temporal nature of the velocity gradients induced by laser ablations and their implications for Fz dynamics. While the velocity gradients induced by laser ablation could be characterised as "transient", the actual timescale of the experiment is 2 hours. These manipulations are intended to model the changes in mechanical stress that cells experience during the developmental process of hinge

contraction that occurs over about 6-8 hours. Hence, we don't consider that the timescales are very different.

Also as mentioned in response to Question 2, we found that the T1 transition rate in fact decreases following acute ablation, suggesting that the acute induction of anisotropic stress favours cell sliding over T1 transitions as the primary stress dissipation mechanism.

Overall, our findings indicate that the short-term response to laser ablation results in a rapid redistribution of Fz, similar to the changes observed during normal developmental processes. In Figure 7, we found that in wild-type wings, as two cells move relative to each other in opposite direction, Fz becomes destabilised at the sliding junctions. The change in Fz stability is quantified before and after the sliding event, within approximately 15 minutes (See Figure 7d-k and Supplementary Figure 7a-c). This strongly suggests that destabilisation of Fz due to cell sliding can occur over minutes to hours in normal development. To clarify, we will revise the manuscript to specify the 15-minute time interval used to measure Fz stability changes at the sliding junctions.

See Results section: Acute induction of a tissue velocity gradient results in reduced Fz stability and altered polarity orientation

We tracked sliding junctions shared between two neighbouring rows of cells moving relative to each other ("sliding junctions") between 24 and 28 hAPF. These sliding junctions were identified as those that underwent a transition of one or both vertices from 4-way to 3-way (Fig. 7d). To assess changes in Fz stability, we measured the stability before and after the transition, within an approximate 15-minute interval.

5. What is the molecular mechanism of Fz destabilization after T1 transition? Is it associated with increase of PTEN (Bardet et al. Dev Cell 2013, for 'typical T1' transition) or detachment of myosin from cellular membrane during T1 (Ikawa et al. Curr. Biol. 2023)?

Thank you for raising this important question regarding the molecular mechanism of Fz destabilisation following T1 transitions.

T1 Transition Rate Does Not Influence Fz Stability

Our findings indicate that Fz destabilisation is not simply a consequence of T1 transition frequency (see response to Question 2). Instead, we demonstrated that the tissue velocity gradient plays a crucial role in modulating Fz dynamics, suggesting that Fz stability is more dependent on the mechanical environment than on the frequency of T1 transitions.

T1 Transition Orientation Does Not Influence Polarity Alignment

Additionally, Ikawa and Sugimura (2018 <https://doi.org/10.1038/s41467-018-05605-7>) showed that defective T1 transition orientation and myosin detachment in wing tissues did not disrupt core pathway planar polarity. This implies that T1 transitions and myosin dynamics may not be the primary factors driving Fz destabilisation. We speculate that, in their study, tissue stress anisotropy remained unaffected, preserving the velocity gradient and thus maintaining core pathway planar polarity.

Similar T1 Transition Mechanisms Yield Different Fz Accumulation and Stability States

In our study, both *dpy-RNAi* wings and the distal regions of wild-type wings exhibited T1 transition events characterised by junction contraction and subsequent elongation, similar to those observed in the proximal regions of wild-type wings (response to Question 2). However,

despite these similar T1 transition mechanisms—such as PTEN enrichment to promote new junction extension—Fz accumulation and stability differed significantly between these regions. This suggests that T1 transitions alone do not govern Fz stability.

Reviewer #2 (Remarks to the Author):

This manuscript describes effects of tissue stress on planar cell polarity (PCP) alignment in the *Drosophila* wing. It builds on (and partially reproduces) early work of Aiguoy et al (2010), who reported that shear reorients PCP during pupal wing development. What this manuscript adds is more insight into the mechanism. By directly examining Fz as a molecular readout for core PCP complexes and using a fluorescent timer construct to assess Fz stability, and correlating effects on Fz with shear stress and the behavior of cells and cell junctions, the authors obtain results that lead them to propose that the main factor reorienting PCP is the disruption of intercellular complexes by shear between neighboring cells. The results provide new insights into how PCP is aligned during development that should be of broad interest. However, there are some issues that need to be addressed.

Major Concerns:

1) There is a major problem with some of the analysis in *dpy* RNAi, as the authors compare distal regions of *dpy* to proximal regions of control, when there are substantial differences between distal and proximal regions. This is not acceptable. This occurs in results presented in Fig. 2 and Fig S1. In Figures 3 and 4 it's not clear whether the comparison between *dpy* results and control results are valid, as it's not clear what regions of the wing were used. This needs to be clarified, and it's essential that the same regions be compared.

We appreciate the reviewer's concern. Our original reasoning was that all regions of the wing are genetically equivalent, and that cell behaviours are primarily determined by the stresses they encounter. However, if you suppose that different regions of the wing respond differently to the same stresses, then problems would occur in comparing different regions within one experiment.

To address this concern, we reanalysed our data to ensure that comparisons are made between equivalent distal regions in both the *dpy-RNAi* wings and the wild-type controls. This approach will eliminate any bias introduced by regional differences. Our re-analysis results indicate that when comparing similar regions between wild-type and *dpy-RNAi* wings, the relationship between tissue velocity gradient and Fz stability remains consistent. In particular, the distal wild-type region, characterised by a higher tissue stress anisotropy, exhibits lower Fz stability compared to the distal *dpy-RNAi* wings, which have a reduced stress anisotropy (Fig. 2h,k).

We modified Figure 2 accordingly to display results obtained from (1) comparison between severed wings and control wings of the same region and (2) comparison between distal *dpy-RNAi* and control wings of the same region.

We have accordingly modified the results section "Manipulation of tissue stress affects Fz PD-polarity coordination and stability" in the revised manuscript.

New Figure 2

Figure 2: Manipulation of tissue stress affects Fz PD-polarity alignment and stability.

2) In Fig 6b, the authors report rotational cell trajectories in *dpy* RNAi, which they call surprising, but they never show us control wings at this age. The wild-type control at this age also needs to be analyzed here.

We appreciate the reviewer's concern about the necessity of appropriate controls for our analysis of cell trajectories in *dpy*-RNAi wings.

The ablation experiments (denoted as PD-shear regions) in our study were conducted specifically on *dpy-RNAi* wings (Fig. 6b,c). The control for this experiment was unablated *dpy-RNAi* wings in the same region. This approach allows us to isolate the effects of the ablation on cell behaviour in the context of *dpy-RNAi* treatment, ensuring that any observed differences are attributable to the ablation rather than due to knocking down of Dumpy.

We agree that examining whether rotational flow exists in wild-type wings would be interesting and we plan to explore this in future studies. For now, our primary objective is to elucidate the specific impacts of acute induction of velocity gradient on PCP alignment and cell behaviour. We will revise the manuscript to clearly state that both control and ablated regions were conducted on the same region and developmental stage in *dpy-RNAi* wings, providing a rationale for this choice. This will be reflected in both the figure legend and the methods section.

See Results section: Acute induction of a tissue velocity gradient results in reduced Fz stability and altered polarity orientation

We then performed laser ablation to induce tissue shearing along the PD-axis in *dpy-RNAi* wings, thereby increasing the tissue velocity_x gradient with respect to the AP-axis (PD-shear) from 16-18 hAPF (Fig. 6c,d and Video 7).

Minor issues:

1) It would be helpful if the authors could relate their analysis of cell flows and Fz orientation in pupal wings to PCP patterns in adult wings, eg describe or refer to analysis of the PCP pattern in *dpy RNAi*.

The *dpy-RNAi* adult wings are not flat due to improper inflation (see image on the right), making it technically difficult to determine the orientation of the adult wing hairs accurately.

While direct analysis of PCP patterns in *dpy-RNAi* adult wings is challenging, we provided wing prehair (trichomes) pattern of *dpy-RNAi* at pupal stage (Fig. 2i). The adult wing hairs are derived from prehairsthat emerge during pupal stage (30-36 hAPF), and there is long-standing evidence that adult hairs broadly reflect the polarity established in the pupa (see Wong & Adler, 1993, 10.1083/jcb.123.1.209).

2) In Fig. 4j, it would be more meaningful if the authors compared Fz stability of horizontal versus proximal junctions, as they did in Fig. 3.

We have reanalysed the data in old Fig. 4j to compare Fz stability at horizontal versus vertical junctions in severed wings (new Fig. 4i). The results showed no statistically significant difference in Fz stability between horizontal and vertical junctions in severed wings, suggesting that the reduced tissue flow gradient leads to loss of PD-polarity coordination in severed wings, which is consistent with our finding on *dpy-RNAi* wings.

See Results section: Anisotropic tissue stress results in a gradient of epithelial flow

Interestingly, severed wings, characterised by a lower tissue velocity_x gradient, exhibited a significantly higher stable proportion of Fz when compared to intact, non-detached control wings (Fig. 4h). Similar to *dpy-RNAi* wings, the relative stable amount of Fz between horizontal

and vertical junctions in severed wings did not show significant differences at 32 hAPF, suggesting reduced tissue velocity gradient leads to loss of PD-polarity coordination in severed wings (Fig. 4i).

(i) Quantification of relative amount of stable Fz on horizontal and vertical junctions for both control and severed wings at 32 hAPF. n = 9–16 wings. Error bars are SEM. Paired t-test, ***P = 0.0005; ns, not significantly different.

3) In Figure 5, the authors describe an experiment that seems not to have worked, so it's not clear that it makes sense to include it. I'm referring to the AP-shear experiment, which didn't show the anticipated differences between horizontal and vertical junctions, which the authors then explain by saying maybe the velocity gradient was too small. If this is the case, then perhaps they should be devising an alternative experiment?

We appreciate the reviewer's feedback on the AP-shear experiment presented in Figure 5. Including this experiment highlights our methodological rigor and transparency in exploring the phenomenon through various experimental setups. While we considered moving it to the supplementary materials, we feel it's important for readers to see our comprehensive exploration and the challenges we faced. The failure of the AP-shear experiment to reorient Fz polarity prompted us to develop an alternative approach to investigate whether a velocity gradient could achieve this, as illustrated in Figure 6. We also note that we did see a change in overall Fz stability in this experiment, so it is not simply negative data.

4) The quantitation for some of the data is unclear. eg How many junctions are represented by the data shown in Fig 3C?

Thanks for pointing this out. We have included the number of junctions analysed for all the data in Figure 3 legend.

5) While the authors have nice explanatory diagrams for some of the data, some of the analysis is complex and could be explained better.

We appreciate your feedback and suggestions for improving the clarity and presentation of our figures. Hopefully by implementing these suggestions, we succeed in enhancing the overall accessibility and interpretability of our findings in the manuscript.

- A summary figure at the end depicting their main conclusion would be helpful.

Thank you for your suggestion. We have now included a new summary figure, added as Figure 8f. We hope that this figure helps to clearly depict the main findings of our study and provides a visual overview of the key conclusions.

(f) Anisotropic stress from wing-hinge contraction induces a PD gradient of cell flow. Shear forces –induced via the gradient of cell flow – destabilise transmembrane core proteins (e.g. Fz and Fmi), hindering their accumulation at junctions parallel to the PD-axis. Consequently, this leads to global polarity alignment along the PD-axis.

- It would also be helpful if the orientation of horizontal versus vertical junctions relative to the pupal wing could be included in an explanatory diagram, as this terminology is not intuitive.

Thank you for pointing this out. We have now included an indication of the axes in the top-left of Figure 3 to indicate the PD- and AP-axes. This should make it clearer that the horizontal junctions in Figure 3d align parallel to the PD-axis, while the vertical junctions align parallel to the AP-axis, as described in the Figure 3 legend.

- In Figures 4 and S5, it would be helpful to have a schematic for the Velocity profile (row and column) in reference to the pupal wing.

We have added an axis bar in Figure 4 and S5 to indicate the PD- and AP-axes of the pupal wing respectively. Hopefully this makes it clear that rows are on the PD axis and columns on the AP axis.

- The dots in Fig 4k are so small that its hard to follow the color scheme.

Thank you for your feedback on the dots in Fig. 4k. While we initially tried enlarging the dots for better visualisation, this led to excessive overlap, reducing readability. Instead, we have adjusted the shape of each dot to be genotype-specific and increased the overall size of the graph to enhance visibility.

- In Figure 1 (& some subsequent figures), the boxes on the wing are a different shape (rectangular) from the image boxes (square), even though they are supposed to be indicating

where the image boxes are from. This should be fixed as its confusing. It would also be nice to have orientation of the image boxes clearly indicated.

Thank you for pointing this out. We have resized the boxes to ensure they are square, consistent with our images. Additionally, we have included an axis bar in all relevant figures to indicate the orientation of the boxes relative to the pupal wing.

- In Fig S4, boxes should be placed on the wing schematic to show us what regions the data being shown comes from.

This has been implemented accordingly.

Reviewer #4 (Remarks to the Author):

Tissue stress orients PCP in both invertebrate and vertebrate epithelia, which is at least partly explained by the slow accumulation of PCP core proteins along nascent junctions (Aigouy et al. Cell, 2010; Wu et al. Curr. Biol., 2016). However, the mechanistic details of this process remain unclear. Tan and Strutt addressed the question by examining the formation of Fz polarity in *Drosophila* pupal wing. They demonstrated that tissue stress anisotropy inversely correlates with Fz stability. Moreover, they observed a correlation between tissue stress anisotropy and differential velocity along cell rows. The acute induction of differential cell flow is sufficient to reorient the Fz polarity in *dpy* RNAi wings. Finally, they showed that Fz and Fmi exhibit lower stability along sliding junctions, which result from differential cell flow. Overall, this study represents an important addition to the field and is worthy of publication after addressing the points below.

Major points

1. Fig. 2. Data acquired in different regions of the wing are compared between control and *dpy* RNAi wings.

We appreciate the reviewer's concern and agree that comparing different regions of the wing between experimental and control groups could introduce confounding variables due to inherent regional differences.

We have reanalysed our data to ensure that comparisons are made between equivalent regions in both the *dpy*-RNAi wings and the wild-type controls. This will eliminate any bias introduced by regional differences. Our re-analysis results indicate that when comparing similar regions between wild-type and *dpy*-RNAi wings, the relationship between tissue velocity gradient and Fz stability remains consistent. In particular, the distal wild-type region, characterised by a higher tissue stress anisotropy, exhibits lower Fz stability compared to the distal *dpy*-RNAi wings, which have a reduced stress anisotropy (Fig. 2h,k).

(See also our response to Question 1 by Reviewer 2)

2. I do not understand why Fz accumulates on newly formed horizontal junctions but not on nearby existing horizontal junctions in Fig. S3, given that both are exposed to similar magnitudes of tissue stress and cell flow gradients at the time of observation. Does the majority of nearby existing horizontal junctions undergo sliding?

We appreciate the reviewer's query regarding the accumulation of Fz on newly formed versus existing horizontal junctions in Figure S3.

Based on our average measurement of the tissue velocity_x gradient, it appears that all horizontal junctions are subjected to a similar velocity gradient from 28-30 hAPF. However, this measurement represents an average across all cells within the imaged region, smoothing out any heterogeneity in the velocity gradients experienced by local cluster of cells. To better assess if horizontal junctions experience varying levels of tissue stress and cell flow gradients, we have now performed a coarse-grain analysis by partitioning the imaged region into smaller areas (approximately 2 to 4 cells per grid) and measured the velocity gradient in each.

The heatmap of coarse-grain velocity_x gradients reveals local variability in the magnitude of velocity gradient experienced by cells across different regions from 28-30 hAPF (New Supplementary Fig. 4d). This variability explains why some cells undergo sliding while others do not.

Additionally, it is possible that newly formed junctions after T1 transitions are “immune to shearing” because the process of T1 transitions acts as a stress relaxation mechanism, reducing the need for further sliding.

See Results section: Anisotropic tissue stress results in a gradient of epithelial flow

To assess the variability in velocity gradients experienced by individual cells, we performed a local velocity_x gradient analysis (referred to as coarse-grain velocity gradient) (Supplementary Fig. 4a-b and see Materials and Methods). The heatmaps of coarse-grain velocity gradients revealed local variability in the magnitude of the velocity_x gradient experienced by cells across different regions (Supplementary Fig. 4d).

(d) Quantification of coarse-grain average velocity_x gradient in the WT proximal region wings from 24-34 hAPF. The average magnitude of velocity_x gradient for each cluster is colour-coded, with pink representing high velocity_x gradients and blue representing low velocity_x gradients.

3. Fig. 4e–g. The difference in the velocity gradient between the distal part of the control wing and the *dpy* RNAi wing is relatively small; yet, the Fz polarity at 32 h APF is severely disrupted in the *dpy* RNAi wing. Please discuss this point.

We appreciate the reviewer's observation and comment regarding Figure 4e–g (new Figure 4e-f), which compares the velocity gradient between the distal region of wild-type wings and *dpy*-RNAi wings. As noted in our response to Reviewer 2, Question 1, we observed that at 24 hAPF, the distal region of wild-type wings does exhibit a higher velocity gradient compared to

the same region in *dpy-RNAi* wings (Fig. 4f), albeit from 28 hAPF onwards, this difference is no longer significant (Fig. 4f).

Furthermore, when comparing tissue velocity gradients at the even earlier stage of 20-22 hAPF, we found again that the distal region of wild-type wings shows a significantly higher velocity gradient than *dpy-RNAi* wings (Supplementary Figure 7e'). Therefore, during earlier stages, the consistently higher velocity gradient in distal control wings most likely contributes to the less severe Fz polarity phenotype observed by 32 hAPF, compared to *dpy-RNAi* wings.

See Discussion section

With regard to our model that cell flows orient Fz polarity, it is interesting to note that even wild-type wing regions with low flows can achieve normal polarity. For instance, the distal regions of wild-type wings show only a similar low gradient of tissue flow to *dpy-RNAi* wings between 24-32 hAPF, yet wild-type wings exhibit PD-oriented Fz polarity, whereas *dpy-RNAi* wings display a swirl polarity phenotype. The most obvious explanation for this, is that in the wild-type wing, higher cell flows in other regions of the wing help to set up PD-oriented polarity which is then transmitted to the distal wing via normal mechanisms of cell-cell propagation of polarity⁹. Furthermore, we find that the distal region of wild-type wings experiences higher velocity gradients from 20-22 hAPF, compared to *dpy-RNAi* wings (Supplementary Fig. 7e-e'). Consequently, these higher velocity gradients during earlier stages may contribute to the PD-oriented Fz polarity observed later in development.

(e) Analysis of tissue flow patterns in the distal region of control WT wings and *dpy-RNAi* wings from 20–22 hAPF. The velocity profile plot illustrates the average velocity of each row of cells. (e') Quantification of average tissue velocity_x gradient of the distal region of control WT wings and *dpy-RNAi* wings from 20–22 hAPF. n = 4 wings. Unpaired t-test, comparing WT distal wings and *dpy-RNAi* wings. **P = 0.0033. Dot indicates mean per wing, error bars are SEM.

4. Tracking the Fz polarity and stability in the same cells would provide further support for the authors' claims. Such analysis could clarify how significantly the mechanism proposed here contributes to the formation of Fz polarity.

In general, it is difficult to get statistically meaningful data from single cell analysis, given the noise inherent in the system. However, in response to the reviewer's comment we took the following approach:

To evaluate the local correlation between velocity gradient and Fz polarity/stability, we computed the coarse-grain velocity gradient and change in Fz stability in clusters of cells. Our analysis revealed that clusters of cells exposed to higher velocity gradients showed a greater reduction in Fz stability on horizontal junctions, while those with lower velocity gradients exhibited less reduction in Fz stability (Supplementary Figure 4g with a $R^2 = 0.5935$), suggesting a correlation between cell flow gradients and Fz stability at junctions parallel to the flow direction. In contrast, Fz stability on junctions perpendicular to the flow direction varied independently of the magnitude of tissue velocity gradients in other regions (Supplementary Figure 4g' with a $R^2 = 0.0118$).

See Results section: Anisotropic tissue stress results in a gradient of epithelial flow

To examine how tissue velocity gradients locally affect Fz stability, we assessed changes in Fz stability across all horizontal junctions in proximal wild-type wings at the stage with highest gradient of cell flow (24- 26 hAPF). First, we clustered the cells into groups and computed the coarse-grain velocity gradient experience by each cluster during this time period. We then determined the change in Fz stability within each cluster from 24-26 hAPF. Our analysis revealed that clusters of cells exposed to higher gradient of cell flow exhibited a greater reduction in Fz stability on horizontal junctions and vice versa, suggesting a correlation between the gradient of cell flow and Fz stability at junctions that are parallel to the flow direction (Supplementary Fig. 4g). In contrast, Fz stability on junctions perpendicular to the flow direction varied independently of the magnitude of tissue velocity gradients in other regions (Supplementary Fig. 4g'). Collectively, these findings suggest that tissue velocity gradients affect the turnover of Fz, particularly on junctions that are parallel to the direction of flow.

New Supplementary Figure 4g-g'

(g,g') Correlation between coarse-grain average velocity_x gradient and percentage of change in Fz stability in the (g) horizontal and (g') vertical junctions of WT wings from 24-26 hAPF. Each dot represents a cluster. R2 indicates the coefficient of determination.

5. Fig. 4k. Could you please enlarge the plot for better visibility? Are the velocity gradient values plotted in Fig. 4k consistent with those shown in Fig. 4e–g?

In the revised manuscript, we have resized Figure 4k (new Figure 4j) within the space available to enhance clarity. Regarding the velocity gradient values, they are the same as those in Fig. 4f (old Fig. 4e-g) plotted against Fz stability. The values were derived using the same analysis approach, ensuring comparability across the figures.

6. Fig. 7 and Fig. S7. Does manipulating cell adhesion strength affect Fz stability and polarity?

We appreciate the reviewer's interest in the potential impact of manipulating cell adhesion strength on Fz stability and polarity.

To address this, we have carried out experiments in which we alter E-cadherin dynamics. Previous studies have suggested that E-cadherin membrane expression regulates adhesion strength, where higher E-cadherin expression correlates with stronger adhesion, and vice versa (e.g. Chu et al., 2004, <https://doi.org/10.1083/jcb.200403043>). Additionally, Iyer et al (2019, <https://doi.org/10.1016/j.cub.2019.01.021>) demonstrated that wings with increased adhesion complex turnover, i.e. in *p120ctn* mutant wings, undergo faster cell shape changes, likely due to weakened cell-cell adhesion. Thus, use of *p120ctn* mutants provides a route to altering cell adhesion strength by modulating E-cadherin dynamics.

More generally, we agree with the proposition that altering adhesion strength could influence Fz stability and polarity, for instance by affecting tissue flow during processes such as cell sliding. For example, stronger adhesion might restrict cell sliding, promoting Fz stabilisation, whereas weaker adhesion could enhance cell movement and potentially destabilise Fz. Alternatively, cell adhesion strength might directly affect Fz stability, for instance by stabilising cell-cell contacts independent of tissue flow.

To test these ideas, we first examined the effect on tissue flow pattern and Fz stability of increasing adhesion complex turnover and thus putatively decreasing adhesion strength in *p120ctn* mutants. Interestingly, we observed that *p120ctn* mutant wings exhibited significantly lower tissue velocity gradients compared to wild-type wings from 24 to 34 hAPF (Fig. 8a-b). We then measured Fz stability in *p120ctn* mutant wings and compared them to control wings in the same proximal region at 28 hAPF. Using FRAP, we found that Fz::EGFP showed significantly greater stability in *p120ctn* mutant wings compared to wild-type control wings (Fig. 8c). Thus, contrary to our initial hypothesis, this result suggests that reduced cell adhesion enhances Fz stability rather than destabilising it – presumably due to the reduced cell flows in *p120ctn* wings. However, comparing local Fz polarity coordination, the *p120ctn* mutant wings also exhibited significantly lower local polarity coordination along the PD-axis (with higher polarity angle variance) as compared to the control wild-type wings at 32 hAPF (Fig. 8d-d'). Overall, we think that reduced adhesion weakens the transmission of tissue stress, resulting in lower tissue flow gradients and consequently increased Fz stability, but also poorer alignment of Fz polarity with the PD axis. This finding supports our model, which posits that tissue velocity gradients influence Fz stability and polarity.

Given the deduced role of tissue velocity gradients in regulating Fz stability, it is important to assess the effect of cell adhesion strength on Fz stability without the confounding influence of tissue stress anisotropy and cell flow gradients. To address this, we used wing-hinge severing to eliminate tissue stress anisotropy and gradients of cell flow.

Notably, Iyer et al (2019) reported that wild-type tissues with reduced stress anisotropy exhibit higher E-cadherin stability suggesting stronger adhesion, so severed control wings represent a putative higher adhesion state with reduced cell flows. Conversely, severed *p120ctn* mutant wings should exhibit lower E-cadherin stability and adhesion, with reduced cell flows. Interestingly, we found no significant difference in Fz::EGFP stability between severed control wings and severed *p120ctn* mutant wings (Fig. 8e), despite the differences in E-cadherin turnover. This suggests that in the absence of cell flow gradients, E-cadherin dynamics do not influence Fz stability.

(We chose wing hinge severing as a method to eliminate tissue velocity gradient over Dumpy knockdown due to time constraints in stock generation.)

These new results are now reported in the final section of the Results (“Relative cell movement leads to a reduction in Fz junctional density and stability”) and new Figure 8a-e.

Figure 8: Manipulating E-Cadherin dynamics does not affect Fz stability

7. p. 12. Ref. 51 and 52 consider active cell rearrangement as a mechanism for inducing tissue shear; however, the cell rearrangement during the developmental stage studied in this manuscript is passively induced as a means to relax tissue stretch. Additional factors, such as the stiffness of veins and wing margin as boundary conditions, as well as spatial variations in cell mechanical/geometrical properties, can be involved. The link between tissue stress and cell flow, therefore, needs to be discussed more carefully. I listed some potentially relevant literature below: Ray et al., Cell, 2015 (Ref. 50), which conducts numerical simulations with different tensions at the vein-intervein boundary; Ishimoto and Sugimura. J Theor. Biol., 2017, regarding the inference of wing margin stiffness; Claussen et al., arXiv, 2024, which describes numerical simulations of patterned flow. I believe there are many more relevant studies.

We thank reviewer for pointing this out. We have now added a little more discussion about the link between tissue stress and cell flow. However, it is beyond the scope of this research manuscript to provide an extensive review of a large area. Furthermore, as the reviewer will appreciate it has been known for over four decades (Gubb & García-Bellido 1982 PMID:6809878) that the final polarity pattern is negligibly affected by changes in the wing margin (e.g. scalloping mutants) or the pattern of venation. To quote the original reference: “if polarity determination occurs during a final stage in development, i.e. within the puparium, it does not take place with reference to the position of the normal wing margins or the distribution

pattern of chaetae or vein tissue". Thus, it seems that the primary influence on polarity is the direction of anisotropic tissue tension provided by hinge contraction, with only subtle influences from the positions of veins and local wing margin attachments to the overlying cuticle. We further suspect that any potential influence of e.g. veins may be masked by the cell-cell coupling of polarity across the wing, which provides a local averaging mechanism that smooths out local perturbations in polarity

See Results section: Anisotropic tissue stress results in a gradient of epithelial flow

Theoretical and numerical simulation studies have shown that gradient flow patterns could be produced by active stress and shear via cell rearrangement^{51,52}. However, in *Drosophila* pupal wing tissue, a gradient of cell flow may serve as a stress-dissipation mechanism to relax tissue under tension. This gradient could result from variations in the wing's geometrical and structural properties, such as the significantly stiffer wing margin⁵¹.

8. Methods. The authors first determined the row of cells based on average cell height and then computed the velocity for each cell row. Please describe how cells undergoing T1 or T2 transitions are treated. I wonder why they did not quantify the velocity field without separating cells into rows and calculate the velocity gradient from the velocity field.

Thank you for your insightful comment. We understand your concerns regarding the treatment of cells undergoing T1 or T2 transitions and the methodology for velocity calculation. Below is our detailed response:

Treatment of T1 and T2 Transitions: We have explicitly considered how cells undergoing T1 (neighbour exchange) and T2 (cell extrusion) transitions were handled during the velocity analysis. All cells, including those undergoing T1 and T2 transitions, were included in the velocity calculations. These transitions generally involve localised cell rearrangements, and their contributions to the local velocity field are captured through our cell-to-grid-based velocity computation. By averaging the velocities of all cells within each grid, we minimise the effects of these transitions on the overall analysis.

The primary reason for choosing this approach was our initial observation that different rows of cells exhibit distinct velocities. By separating the cells into rows, we were able to capture and quantify this variability effectively. We opted to calculate the velocity gradients between rows and columns of cells because the other velocity gradient tensor component values ($\frac{\partial v_x}{\partial x}$ and $\frac{\partial v_y}{\partial y}$) were generally very low, rendering them negligible in a global analysis.

We have further improved the method by introducing a cell-to-grid based analysis by partitioning the imaged region into clusters of cells, adapted according to previous published method (Smutny et al., 2017 <https://doi.org:10.1038/ncb3492>). This refinement makes the analysis more sensitive and robust, taking into account of variations in velocity gradients across the entire imaged region. We have reanalysed all our data using this updated method and revised the manuscript accordingly. Overall, we found that the results were not statistically different from those obtained using our original method.

We have added the improved method into our Materials and Methods section under "Quantification of tissue velocity gradient"

Minor points

1. Methods. Please provide the definition of the polarity magnitude quantified using QuantifyPolarity (Ref. 44).

Thank you for this. We have added in an explanation for the definition of polarity magnitude in the Materials and Methods section under "Cell polarity, circular weighted histogram and eccentricity analysis".

See Materials and Methods section: Cell polarity, circular weighted histogram and eccentricity analysis

The polarity magnitude and angle of individual cells are calculated using a Principal Component Analysis (PCA)-based method, which identifies the direction of the greatest variance in weighted intensities from the cell's centroid. When proteins are evenly distributed across cell junctions, the cell is unpolarised. However, if proteins are asymmetrically distributed to opposite junctions, known as 'bipolarity,' the cell becomes polarised with a higher polarity magnitude. The polarity angle is defined as the axis with the greatest asymmetry.

2. Please set the minimum value of the y-axis to zero to accurately represent the data, except for the stable fraction of Fz, where cells should maintain a certain value for junctional localization.

Thank you for the suggestion regarding setting the y-axis to zero on the graphs. There are a number of cases where we can see that this makes sense and have implemented appropriate changes. However, we do not believe it makes sense to do this when considering cell polarity as measured by the PCA method. In this case, adjusting the y-axis in such a way would distort the interpretation of the data. The key range of interest for cell polarity in both Sqh and Fz data is between 0.1 and 0.4, which represents the biologically relevant variation in polarity measurements. Setting the y-axis to zero would artificially compress this range, making it difficult to visualise and interpret subtle changes in polarity.

In response to your feedback, we have standardised all polarity graphs to focus on the range of 0.1 to 0.4 (i.e. a constant y-axis scale). This adjustment ensures consistency across our figures, highlighting the meaningful variations within the biologically relevant range.

3. Fig. 3a. Yellow and blue in the second schematic in the second row are swapped.

Thank you for spotting this. We have changed this accordingly.

4. Fig. 3f, g and other figures. Please specify the region of the wing from which the data were collected.

Thanks for this. We have specified the wing regions for all relevant figures.

5. Fig. 4 and 5. Why is the AP-axis inverted?

We believe that this orientation offers a more intuitive visualisation of the velocity profiles. By showing lower velocity near the wing margin at the top and higher velocity toward the bottom, we can clearly present an ascending velocity gradient. Importantly, the orientation of the axis does not impact the overall interpretation of our findings.

6. Fig. 4k. R^2 is shown as 0.7923 in the figure, whereas it is reported as 0.7668 in the main text. Please clarify.

We apologise for any confusion caused by this error and appreciate the reviewer bringing it to our attention. We have revised the main text to the correct R^2 value of 0.7923 (now Fig. 4j).

7. Fig. 5 and Fig. 6. Line ablation halts tissue flow rather than promoting it in the *Drosophila* embryo, as reported by Collinet et al. (Nat. Cell Biol., 2015). Please discuss this point.

We appreciate the reviewer highlighting the comparison between our results in Fig. 5 and Fig. 6 and the findings by Collinet et al. (2015) regarding the effects of line ablation on tissue flow in the *Drosophila* embryo.

To clarify, our results are consistent with Collinet et al. for the first 30 minutes post-ablation, during which both studies observed halted tissue flow and primarily cell elongation in response to wound contraction. However, while Collinet et al. did not monitor cell dynamics beyond this 30-minute period, we extended our observations to 2 hours post-ablation. As the wound continued to contract after the initial 30 minutes, we observed a resumption of cell flow, with cells migrating toward the wound margin, driven by ectopic contraction forces. This extended timeframe allowed us to capture dynamic tissue responses that occur later in the wound healing process, which were not addressed in Collinet et al.'s study. Thus, the discrepancy is not in the early phase of response but rather in the longer-term observations of tissue behaviour post-ablation.

Additionally, it is speculated that laser ablation in the *Drosophila* embryo led to tissue fusion with the vitelline membrane, which could prevent tissue flow. This mechanism differs from that in the *Drosophila* pupal wing, where the wing epithelial tissue is located farther from the cuticle.

8. Fig. S1e. Labeling is missing.

Thanks for pointing this out. We have added proximal and distal labelling.

9. Ref. 56 is not complete.

Thank you for noticing this. We have added the DOI for Ref. 56 accordingly.

Response to reviewers

Reviewer #1 (Remarks to the Author):

The revised manuscript shows significant improvement, with the addition of some excellent figures and discussions. The authors' claims are now clearer, and their responses to my comments are satisfactory. I agree that exploring the molecular mechanism is beyond the scope of this work and should be addressed in future studies. I now recommend the manuscript for publication in Nature Communications.

I would like the author to clarify the following points before the publication.

1. I am still confused about the relationship between sliding and shear. In your laser ablation experiment, do cell membranes slide against each other? The observed shear could be attributed to cell shape changes with minimal cell membrane sliding.

Thank you for raising this point. Upon induction of tissue shearing we observe relative cell movement; however, we are unable to confirm if during this process the cell membranes slide against each other. This is a question we have thought about, and we think that to answer it we would need to label membranes in adjacent cells in different colours and track them over time. Such experiments would require generation of new fly strains and are something that we would like to pursue in future studies

Regarding cell shape changes, we think that the reviewer is suggesting that the observed cell movement could be mediated by a process of junctional remodelling that progressively changes cell shapes, at least in the sense that there are changes in the relative positions of vertices (possibly without any significant change in overall cell shape). We actually think that this is quite plausible as a mechanism and so instead of membranes sliding, junctions extend and shrink to mediate relative cell movement. This is in fact how we describe the process in the text and Figure 7. However, we don't think that this rules out shear between membranes of adjacent cells during relative cell movement, as such shear could be the trigger for junction remodelling.

Our current experiments do not clearly distinguish between these possibilities, hence our care in the Results and Discussion to avoid implying we fully understand the mechanisms.

Minor points:

1. Some figures (plots and histograms) are small, making it difficult to see the characters.

Thank you for pointing out that some figures (plots and histograms) are quite small. We have adjusted figure sizes to improve visibility and hopefully ensure that all characters and details are clear e.g. all text is ≥ 6 pt in size.

2. L375: Fig. 4a,b, 4f should be changed to Fig. 4a,b,f.

The text has been revised to replace "Fig. 4a,b, 4f" with "Fig. 4a,b,f"

3. L399: A period is missing.

Thank you for catching this error. We have added the missing period.

4. L400: The figure number is missing.

We have corrected the text to include the missing figure number.

Reviewer #2 (Remarks to the Author):

The authors revisions have addressed the concerns raised - it's an important addition to the field.

We thank the reviewer for their help in improving the previous version of the manuscript.

Reviewer #3 (Remarks to the Author):

Reviewer #4 (Remarks to the Author):

The authors have greatly improved the manuscript by performing a series of new experiments and data analyses. I agree with the publication of this study.

Below are some minor comments that I hope will be helpful:

- Fig. 1c. Replace v_{maj} with v_{max} .

Thank you for your suggestion. We think the reviewer is asking us to pair 'min' with 'max', but as indicated in Fig.S1, the nomenclature we are using is 'minor' vs 'major' so the correct abbreviation for 'major' is 'maj'. We have therefore opted to continue using the term ' v_{maj} ' in line with the conventional terminology established in the published method we referenced.

- Fig. 8. I would suggest replacing "dynamics" with "turnover," as the former covers wider aspects, such as lateral clustering.

Thank you for the suggestion. We have updated the figure title and corresponding text to replace "dynamics" with "turnover."

- Fig. S3a. The remodeling junction in the second panel of the top row is not a 4-way vertex. I would suggest using the same labeling as in Fig. 3.

We appreciate your observation regarding the junction in the second panel of the top row. We have corrected the figure to ensure it accurately represents a 4-way vertex and have updated the labelling to align with that used in Fig. 3 for consistency.

- Fig. S4g and g'. These panels demonstrate the correlation between the tissue velocity gradient and Fz stability. They are worth including in the main text if space allows.

Thank you for the suggestion. Due to space constraints, we have decided to keep these panels in the supplementary material while maintaining their relevance for interested readers.

- Line 346–350. The ref. 51 and 52 in the original manuscript (Sato et al., 2015; Popovic et al., 2017) are missing from the bibliography of the revised manuscript. It seems that there are inconsistencies between the main text and the bibliography.

We apologise for the oversight. The references to Sato et al., 2015, and Popovic et al., 2017, were inadvertently omitted from the bibliography. These have been re-added and cross-checked to ensure consistency with the citations in the main text.

- Line 400. The figure number is missing.

Thank you for pointing this out. We have corrected the text to include the missing figure number.

Below are some discussions that could be informative for the authors' future studies:

- The MT growth polarity varies along the PD axis (Harumoto et al., 2010), similar to the spatial variation in the tissue flow gradient that the authors demonstrated in this study. I wonder if these two are regulated by a common mechanism. If so, are they chemically prepatterned or a consequence of mechanotransduction?

- As the authors assumed in their response to reviewer 1, tissue stress anisotropy was unaffected by RNAi of *aip1* or *cofilin* (Fig. S4 in the reference).

We agree that the spatial variation in tissue flow gradient, similar to the variation in MT growth polarity along the PD axis (Harumoto et al., 2010), suggests an intriguing potential relationship between these two phenomena. Exploring whether they share a common regulatory mechanism, and whether they are prepatterned or a consequence of mechanotransduction, would indeed provide valuable insights. This is an exciting direction for future work, and we plan to investigate it further.